



**Sixty years of radiocarbon dioxide measurements at Wellington, New**
**Zealand 1954 – 2014**
Jocelyn C. Turnbull[1,2,*] Sara E. Mikaloff Fletcher[3], India Ansell[1], Gordon Brailsford[3],
Rowena Moss[3], Margaret Norris[1], Kay Steinkamp[3]
[1]GNS Science, Rafter Radiocarbon Laboratory, Lower Hutt, New Zealand
[2]CIRES, University of Colorado at Boulder, Boulder, Colorado, USA
[3]NIWA, Wellington, New Zealand
* contact author: j.turnbull@gns.cri.nz

## 1. Abstract

We present 60 years of $\Delta^{14}CO_2$ measurements from Wellington, New Zealand (41°S,
175°E). The record has been extended and fully revised. New measurements have been
used to evaluate the existing record and to replace original measurements where
warranted. This is the earliest atmospheric $\Delta^{14}CO_2$ record and records the rise of the $^{14}C$
"bomb spike", the subsequent decline in $\Delta^{14}CO_2$ as bomb $^{14}C$ moved throughout the
carbon cycle and increasing fossil fuel $CO_2$ emissions further decreased atmospheric
$\Delta^{14}CO_2$. The initially large seasonal cycle in the 1960s reduces in amplitude and
eventually reverses in phase, resulting in a small seasonal cycle of about 2 ‰ in the
2000s. The seasonal cycle at Wellington is dominated by the seasonality of cross-
tropopause transport, and differs slightly from that at Cape Grim, Australia, which is
influenced by anthropogenic sources in winter. $\Delta^{14}CO_2$ at Cape Grim and Wellington
show very similar trends, with significant differences only during periods of known
measurement uncertainty. In contrast, Northern Hemisphere clean air sites show a higher
and earlier bomb $^{14}C$ peak, consistent with a 1.4-year interhemispheric exchange time.
From the 1970s until the early 2000s, the Northern and Southern Hemisphere $\Delta^{14}CO_2$
were quite similar, apparently due to the balance of $^{14}C$-free fossil fuel $CO_2$ emissions in
the north and $^{14}C$-depleted ocean upwelling in the south. The Southern Hemisphere sites
show a consistent and marked elevation above the Northern Hemisphere sites since the
early 2000s, which is most likely due to reduced upwelling of $^{14}C$-depleted and carbon-
rich deep waters in the Southern Ocean. This developing $\Delta^{14}CO_2$ interhemispheric
gradient is consistent with recent studies that indicate a reinvigorated Southern Ocean
carbon sink since the mid-2000s, and suggests that upwelling of deep waters plays an
important role in this change.



## 2. Introduction

Measurements of radiocarbon in atmospheric carbon dioxide ($\Delta^{14}CO_2$) have long been
used as a key to understanding the global carbon cycle. The first atmospheric $\Delta^{14}CO_2$
measurements were begun at Wellington, New Zealand in 1954 (Rafter, 1955; Rafter et
al., 1959), aiming to better understand carbon exchange processes (Otago Daily Times,
1957). Northern Hemisphere $\Delta^{14}CO_2$ measurements began a few years later in Norway
(Nydal and Løvseth, 1983) and Austria (Levin et al., 1985).
$^{14}C$ is a cosmogenic nuclide produced naturally in the upper atmosphere through neutron
spallation, exchanges to form $^{14}CO_2$ and then moves throughout the global carbon cycle.
Production is roughly balanced by radioactive decay, which mostly occurs in the carbon-
rich and slowly overturning ocean carbon reservoir and to a lesser extent in the faster
cycling terrestrial carbon reservoir. The perturbations to $\Delta^{14}CO_2$ from atmospheric
nuclear weapons testing in the mid-20$^{th}$ century and additions of $^{14}C$-free $CO_2$ from fossil
fuel burning have both provided tools to investigate $CO_2$ sources and sinks.
Penetration of bomb-$^{14}C$ into the oceans has been used to understand ocean carbon uptake
processes (Oeschger et al., 1975; Broecker et al., 1985; Key et al., 2004; Sweeney et al.,
2007; Naegler et al., 2006). Terrestrial biosphere carbon residence times and exchange
processes have also been widely investigated using bomb-$^{14}C$ (e.g. Trumbore et al., 2000;
Naegler et al., 2009). Stratospheric residence times, cross-tropopause transport and
interhemispheric exchange can also be examined with atmospheric $\Delta^{14}CO_2$ observations
(Kanu et al., 2015; Kjellström et al., 2000).
The Suess Effect, the decrease in atmospheric $\Delta^{14}CO_2$ due to the addition of $^{14}C$-free
fossil fuel $CO_2$, was first identified in 1955 (Suess, 1955). It has subsequently been
refined (Levin et al., 2003; Turnbull et al., 2006) and used to investigate fossil fuel $CO_2$
additions at various scales (e.g. Miller et al., 2012; Turnbull et al., 2009; Lopez et al.,
Turnbull et al., 2015; Djuricin et al., 2010).
The full atmospheric $^{14}C$ budget has been investigated using long term $\Delta^{14}CO_2$ records in
conjunction with atmospheric transport models (Randerson et al., 2002; Caldiera et al.,
1998; Levin et al., 2010; Turnbull et al., 2009; Naegler et al., 2006). These have shown
changing controls on $\Delta^{14}CO_2$ through time. Prior to nuclear weapons testing, natural
cosmogenic production added $^{14}C$ in the upper atmosphere, which reacted to $CO_2$ and
moved throughout the atmosphere and the carbon cycle. The short carbon residence time
in the biosphere meant that biospheric exchange processes had only a small influence on
$\Delta^{14}CO_2$, whereas the ocean exerted a stronger influence due to radioactive decay during
its much longer (and temporally varying) turnover time. The addition of bomb $^{14}C$ in the
1950s and 1960s almost doubled the atmospheric $^{14}C$ content. This meant that both the
ocean and biosphere were now very $^{14}C$-poor relative to the atmosphere. As the bomb-
$^{14}C$ was distributed throughout the carbon cycle, this impact weakened, and by the 1990s,
the additions of fossil fuel $CO_2$ became dominant.
The long-term $\Delta^{14}CO_2$ records have been crucial in all of these findings, and the
Wellington $\Delta^{14}CO_2$ record is of especial importance, being the oldest direct atmospheric



trace gas record, even predating the $CO_2$ mole fraction record started at Mauna Loa in
1958 (Keeling, 1961; Keeling and Whorf, 2005). It is the only Southern Hemisphere
record recording the bomb spike.  Several short Southern Hemisphere records do exist
(Meijer et al., 2006; Graven et al., 2012b; Manning et al., 1990; Hua and Barbetti 2013),
and some longer records began in the 1980s (Levin et al., 2010).  Over the more than 60
years of measurement, there have necessarily been changes in how the Wellington
samples are collected and measured.  There are no comparable records during the first 30
years of measurement, so that the data quality has not been independently evaluated.
Comparison with other records since the mid-1980s has suggested that there may be
biases in some parts of the Wellington record (Currie et al., 2011).

Here we present a revised and extended Wellington atmospheric $^{14}CO_2$ record, spanning
60 years from December 1954 to December 2014.  We detail the different sampling,
preparation and measurement techniques used through the record, compare with new tree
ring measurements, discuss revisions to the previously published data and provide a final
dataset with accompanying smooth curve fit.

In the results and discussion, we revisit the key findings that the Wellington $^{14}CO_2$ record
has provided over the years and expand with new findings based on the most recent part
of the record. The most recent publication of this dataset included data to 2005 (Currie et
al., 2011) and showed periods of variability and a seasonal cycle at Wellington that differ
markedly from the independent Cape Grim, Tasmania $^{14}CO_2$ record at a similar southern
latitude (Levin et al., 2010).  Here we add complementary new data to investigate these
differences, fill gaps and extend the record to near-present.  We examine an emerging
interhemispheric gradient in $^{14}CO_2$, which supports evidence of a changing Southern
Ocean carbon sink.  If this emerging gradient is indeed linked to Southern Ocean
processes, it suggests that ocean circulation plays a substantive role in the reinvigoration
of the Southern Ocean carbon sink.
## 3. Methods
Over 60 years of measurement, a number of different sample collection, preparation,
measurement and reporting methods have been used.  In this section, we give an
overview of the various methods and changes through time, and they are summarized in
table 1.  Full details of the sampling methods used through time are provided in the
supplementary material, compiling methodological information documented in previous
reports on the Wellington record (Rafter and Fergusson, 1959; Manning et al., 1990;
Currie et al., 2011) along with methods newly applied in this new extension and
refinement of the dataset.

### 3.1. Sampling sites
Samples from 15 December 1954 – 5 June 1987 were collected at Makara (Lowe, 1974),
on the south-west coast of the North Island of New Zealand (MAK, 41.25°S, 174.69°E,
300 m asl). Samples since 8 July 1988 have been collected at Baring Head (Brailsford et
al., 2012) on the South Coast of the lower North Island and 23 km southeast of Makara
(BHD, 41.41°S, 174.87°E, 80 m asl) (figure 1).  We also discuss tree ring samples
collected from Eastbourne, 12 km north of Baring Head on Wellington Harbour.


### 3.2. Collection methods

#### 3.2.1. NaOH absorption

The primary collection method is static absorption of $CO_2$ into nominally $CO_2$-free 0.5 or
1 M $L^{-1}$ sodium hydroxide (NaOH) solution, which is left exposed to air at the sampling
site providing an integrated sample over a period of ~2 weeks (Rafter, 1955). From
1954-1995, ~ 2 L NaOH solution was exposed to air in a large Pyrex® tray. Since 1995,
high-density polyethylene (HDPE) bottles containing ~200 mL NaOH solution were left
open inside a Stevenson meteorological screen; the depth of the solution in the bottles
remained the same as that in the previously used trays. No significant difference has been
observed between the two methods (Currie et al., 2011). A few early (1954-1970)
samples were collected using different vessels, air pumped through the NaOH (vs.
passive absorption), or NaOH was replaced with barium hydroxide (Rafter, 1955;
Manning et al., 1990). $CO_2$ is extracted from the NaOH solution by acidification followed
by cryogenic distillation (Rafter and Fergusson, 1959; Currie et al., 2011).

#### 3.2.2. Whole air flasks

In this study, we use whole air flask samples collected at Baring Head to supplement
and/or replace NaOH samples. Flasks of whole air are collected by flushing ambient air
through the flask for several minutes then filled to slightly over ambient pressure. Most
flasks were collected during southerly, clean air conditions (Stephens et al., 2013). $CO_2$
is extracted cryogenically (Turnbull et al., 2015). For whole air samples collected from
1984-1993, the extracted $CO_2$ was archived until 2012. We evaluated the quality of this
archived $CO_2$ using two methods. Tubes with major leakage were readily detected by air
present in the tube and were discarded. $\delta^{13}C$ from all the remaining samples was in
agreement with $\delta^{13}C$ measured from separate flasks collected at Baring Head and
measured for $\delta^{13}C$ by Scripps Institution of Oceanography at close to the time of
collection (http://scrippsco2.ucsd.edu/data/nzd). Whole air samples collected since 2013
are analyzed for $\delta^{13}C$ and other trace gases and isotopes at NIWA (Ferretti et al., 2000)
and for the $^{14}CO_2$ measurement, $CO_2$ is extracted from whole air at Rafter Radiocarbon
Laboratory (Turnbull et al., 2015).

#### 3.2.3. Tree rings

When trees photosynthesize, they faithfully record the $^{14}C$ content of ambient $CO_2$ in
their cellulose, the structural component of wood. Annual tree rings therefore provide a
summertime (approximately September – April in the Southern Hemisphere) daytime
average $\Delta^{14}CO_2$. Photosynthetic uptake varies during the daylight hours depending on
factors including growth period, sunlight, and temperature (Bozhinova et al., 2013),
resulting in a somewhat different effective sampling pattern than the 1-2 week NaOH
solution collections. We show in section 3.5.1. that at the Wellington location this
difference is negligible. Note that we assign the mean age of each ring as January 1 of the
year in which growth finished (i.e. the mean age of a ring growing from September –
April), whereas dendrochronologists assign the "ring year" is as the year in which ring
growth started (i.e. the previous year).
We collected cores from three trees close to the Baring Head site. A pine (*Pinus radiata*)
located 10 m from the Baring Head sampling station (figure 1) yielded rings back to 1986



(Norris, 2015). A longer record was obtained from two New Zealand kauri (*Agathis*
*australis*) specimens planted in 1919 and 1920, located 20 m from one another in
Eastbourne, 12 km from Baring Head (figure 1). Kauri is a long-lived hardwood species
that has been widely used in dendrochronology and radiocarbon calibration studies (e.g.
Hogg et al., 2013).
Annual rings were counted from each core. Shifting the Eastbourne record by one year in
either direction moves the $^{14}$C bomb spike maximum out of phase with the NaOH-based
Wellington $\Delta^{14}CO_2$ record (supplementary figure S1), confirming that the ring counts are
correct. For the Baring Head pine, rings go back to only 1986, and we verify them by
comparing with the Eastbourne record. They show an insignificant mean difference of -
$0.4 \pm 0.8$ ‰ (supplementary figure S1).
In practice, it is difficult to ensure that one annual ring is sampled without losing any
material from that ring, and no wood from surrounding rings is included. To evaluate the
potential bias from this source, we measured replicate samples from different cores from
the same tree (Baring Head) or two different trees (Eastbourne, 12 km north of Baring
Head). For samples collected since 1985, all these replicates agree within one standard
deviation (supplementary figure S2). However, for three replicates from Eastbourne in
1963, 1965 and 1971, we see large differences of 9.2, 44.5 and 4.9 ‰, which we attribute
to small differences in sampling of the rings that were magnified by the rapid change in
$\Delta^{14}$C of up to 200 ‰ yr$^{-1}$ during this period. Thus, the tree ring $\Delta^{14}$C values during this
period should be treated with caution.
Cellulose was isolated from whole tree rings by first removing labile organics with
solvent washes, then oxidation to isolate the cellulose from other materials (Norris, 2015;
Hua et al., 2000). The cellulose was combusted and the $CO_2$ purified following standard
methods in the Rafter Radiocarbon Laboratory (Baisden et al., 2013).
### 3.3.  $^{14}$C measurement
Static NaOH samples were measured by conventional decay counting on the $CO_2$ gas
from 1954 – 1995 (Manning et al., 1990; Currie et al., 2011) and these are identified by
their unique "NZ" numbers. All measurements made since 1995, including recent
measurements of flask samples collected in the 1980s and 1990s, were reduced to
graphite, measured by accelerator mass spectrometry (AMS), and are identified by their
unique "NZA" numbers. The LG1 graphitization system was used from 1995 to 2011
(NZA < 50,000) (Lowe et al., 1987), and replaced with the RG20 graphite system in 2011
(NZA > 50,000) (Turnbull et al., 2015). Samples measured by AMS were stored for up
to three years between sample collection and extraction/graphitization/measurement.
For samples collected from 1995 to 2010, an EN Tandem AMS was used for
measurement (NZA < 35,000, Zondervan and Sparks, 1996). Until 2005 (NZA <30,000,
including all previously reported Wellington $^{14}CO_2$ data), only $^{13}$C and $^{14}$C were
measured on the EN Tandem system, so the normalization correction for isotopic
fractionation (Stuiver and Polach, 1977) was performed using an offline isotope ratio
mass spectrometer $\delta^{13}$C value. The data reported from 2005 onwards (NZA > 30,000)



show a reduction in scatter reflecting the addition of online $^{12}$C measurement in the EN
Tandem system in 2005.  This allows direct online correction for isotopic fractionation
that may occur during sample preparation and in the accelerator itself (Zondervan et al.,
2015), and results in improved long-term repeatability.
For all EN Tandem samples, a single large aliquot of extracted $CO_2$ was split into four
separately graphitized and measured targets and the results of all four were averaged. We
have revisited the multi-target averaging, applying a consistent criterion to exclude
outliers and using a weighted mean of the retained measurements (supplementary
material).  This results in differences of up to 5 ‰ relative to the values reported by
Currie et al. (2011).
In 2010, the EN Tandem was replaced with a National Electrostatics Corporation AMS,
dubbed XCAMS (NZA > 34,000). XCAMS measures all three carbon isotopes, such that
the normalization correction is performed using the AMS measured $^{13}$C values
(Zondervan et al., 2015). XCAMS measurements are made on single graphite targets
measured to high precision (Turnbull et al., 2015).

## 3.4. Results format
NaOH samples are collected over a period of typically two weeks, and sometimes much
longer.  We report the date of collection as the average of the start and end dates.  In
cases where the end date was not recorded, we use the start date.  For a few samples, the
sampling dates were not recorded or are ambiguous, and those results have been excluded
from the reported dataset.
Results are reported here as $F^{14}$C (Reimer et al., 2004) and $\Delta^{14}$C (Stuiver and Polach,
1977).  $F^{14}$C is corrected for isotopic fractionation and blank corrected.  We calculated
$F^{14}$C from the original measurement data recorded in our databases, and updated a
handful of records where transcription errors were found.  $\Delta^{14}$C is derived from $F^{14}$C, and
corrected for radioactive decay since the time of collection (Stuiver and Polach, 1977).
$\Delta^{14}$C has been recalculated using the date of collection for all results, resulting in changes
of a few tenths of permil in most $\Delta^{14}$C values relative to those reported by Currie et al.
(2011) and Manning et al. (1990).  Uncertainties are reported based on the counting
statistical uncertainty and for AMS measurements we add an additional error term,
determined from the long-term repeatability of secondary standard materials (Turnbull et
al., 2015).  Samples for which changes have been made relative to the previously
published results are indicated by the quality flag provided in the supplementary dataset.
Where more than one measurement was made for a given date, we report the weighted
mean of all measurements.

## 3.5. Data validation
*3.5.1. Tree ring comparison*
Over the more than 60 years of the Wellington $\Delta^{14}CO_2$ record, there have necessarily
been many changes in methodology, and the tree rings provide a way to validate the full
record, albeit with lower resolution.  Due to the possible sampling biases in the tree rings



(section 3.2.3.), we do not include them in the final updated record, but use them to
validate the existing measurements.
During the rapid $\Delta^{14}CO_2$ change in the early 1960s, there are some differences between
the kauri tree ring and Wellington $\Delta^{14}CO_2$ records. The 1963 and 1964 tree ring samples
are slightly lower than the concurrent $\Delta^{14}CO_2$ samples. The peak $\Delta^{14}CO_2$ measurement
in the tree rings is 30 ‰ lower than the smoothed $\Delta^{14}CO_2$ record, and 100‰ lower than
the two highest $\Delta^{14}CO_2$ measurements in 1965. These differences are likely due to small
errors in sampling of the rings, which will be most apparent during periods of rapid
change.
Prior to 1960 and from the peak of the bomb spike in 1965 until 1990, there is remarkable
agreement between the tree rings and Wellington $\Delta^{14}CO_2$ record, with the wiggles in the
record replicated in both records. And since 2005, there is excellent agreement across all
the different records. Some differences are observed in 1990-1993 and 1995-2005, which
we discuss in the following sections.
*3.5.2. 1990-1993 excursion*
An excursion in the gas counting measurements between 1990 and 1993 has previously
been noted (figures 2, 3) as a deviation from the Cape Grim $\Delta^{14}CO_2$ record (Levin et al.,
2010) during the same period. Cape Grim is at similar latitude, and observes a mixture of
air from the mid-latitude Southern Ocean sector and terrestrial Australia (Law et al.,
2010; Ziehn et al., 2014). The Wellington and Cape Grim records overlap during almost
all other periods (figure 3).
We use archived $CO_2$ from flask samples to evaluate this period of deviation. First, the
recent flask samples collected since 2013 (n=12) agree very well with the NaOH static
samples from the same period (figure 2), indicating that despite the difference in
sampling period for the two methods, flask samples reflect the $\Delta^{14}CO_2$ observed in the
longer-term NaOH static samples. We then selected a subset of archived 1984 - 1992
extracted $CO_2$ samples for measurement, mostly from Southerly wind conditions, but
including a few from other wind conditions. These flask $\Delta^{14}CO_2$ measurements do not
exhibit the excursion seen in the NaOH static samples (figure 2), implying that the
deviation observed in the original NaOH static samples may be a consequence of
sampling, storage or measurement errors. Annual tree rings from both the kauri and pine
follow the flask measurements for this period (figure 2), confirming that the NaOH static
samples are anomalous.
The 1990-1993 period was characterized by major changes in New Zealand science, both
in the organizational structure and personnel. Although we are unable to exactly
reconstruct events at that time, we hypothesize that the NaOH solution preparation was
conducted slightly differently, perhaps omitting the barium chloride precipitation step for
these samples. This would result in contaminating $CO_2$ absorbed on the NaOH before the
solution was prepared, which would result in higher $\Delta^{14}CO_2$ observed in these samples
than in the ambient air. In any case, these values are anomalous and we remove the





original NaOH static sample measurements between 1990 and 1993 and replace them
with the new flask measurements for the same period.
*3.5.3. 1995-2005 variability*
As already discussed in section 3.3, the measurement method was changed from gas
counting to AMS for samples collected in 1995 or thereafter.  During the first ten years of
AMS measurements, the record is much noisier than during any other period (figure 2).
In 2005, online $^{12}$C measurement was added to the AMS system, substantially improving
the measurement accuracy (Zondervan et al., 2015), and the noise in the $\Delta^{14}CO_2$ record
immediately reduced.
The remaining NaOH solution for all samples collected since 1995 has been archived,
and typically only every second sample collected was measured, with the remainder
archived without sampling.  In 2011-2016, we revisited the 1995-2005 period,
remeasuring some samples that had previously been measured and some that had never
been measured for a total of 52 new analyses.
The new measurements on this period do show reduced scatter over the original analyses,
particularly for the period from 1998-2001 where the original analyses appear
anomalously low and in 2002-2003 when the original analyses appear anomalously high.
Yet there remain a number of both low and high outliers in the new measurements.
These are present in both the samples that were remeasured and in those for which this
was the first sample from the bottle.  This suggests that a subset of the archived sample
bottles were either contaminated at the time of collection, or that some bottles were
insufficiently sealed, causing contamination with more recent $CO_2$ during storage.
Comparison with the tree ring measurements and with the Cape Grim record (Levin et al.,
2010) suggest that the measurements during this period may, on average, be biased high
as well as having additional scatter (figure 3). Nonetheless, in the absence of better data,
we retain both the original and remeasured NaOH sample results in the full record.
## 3.6.  Smooth curve fit
In addition to the raw measured $\Delta^{14}CO_2$ values, we calculate a smooth curve fit and
deseasonalized trend from the Wellington $\Delta^{14}C$ and $F^{14}C$ datasets.  The deseasonalized
trend may be more useful than the raw data for aging of recent materials (e.g. Reimer et
al., 2004; Hua et al., 2013). Acknowledging that the 1995-2005 period is variable and
possibly biased in the Wellington record, we also provide in the supplementary material
an alternative mid-latitude Southern Hemisphere smooth curve fit and deseasonalized
trend in which the Wellington data for 1995-2005 has been removed and replaced with
the Cape Grim data for that period (Levin et al., 2010).
Curvefitting is particularly challenging for the $\Delta^{14}CO_2$ record, since (a) there are data
gaps and inconsistent sampling frequency, (b) the growth rate and trend vary dramatically
and (c) the seasonal cycle changes both in magnitude and phase (section 4.2). We chose
to use the ccgvu fitting procedure (Thoning et al., 1989), which uses fast Fourier
transform and low-pass filtering techniques to obtain a smoothed seasonal cycle and long
term trend from atmospheric data.  This technique can readily handle the data gaps and





inconsistent sampling frequency in our record, whereas the other widely used fitting
procedure, seasonal trend decomposition using locally weighted scatter plot smoothing
(STL) requires gap-filling for our dataset (e.g. Pickers et al., 2015). However, ccgvu
assigns a single set of harmonic terms across the full time period, which is inappropriate
in this case of large variation in the seasonal cycle. Thus, we separate the record into five
time periods, chosen as periods when changes in the growth rate, seasonal cycle and data
quality change: 1954-1965, 1966-1979, 1980-1989, 1990-2004, 2005-2014. For each
time period, we use ccgvu with one linear and two harmonic terms and fit residuals are
added back using a low-pass filter with an 80 cutoff in the frequency domain. At each
transition, we overlapped a two-year period and linearly interpolated the two fits across
that two year period to smooth the transitions caused by end effects. The deseasonalized
trend was determined from the full dataset rather than the five time periods, as it does not
include the seasonality and produces the same result in either case.

The mean residual difference between the fitted curve and the measured $\Delta^{14}CO_2$ values is
3.8 ‰, consistent with the typical measurement uncertainty for the full dataset. Further,
the residuals are highest for the early period (1954-1970) at 6 ‰, consistent with the
larger measurement errors at that time of ~6 ‰. The residuals improve as the
measurement errors reduce, such that since 2005, the mean residual is 2 ‰, consistent
with the reported 2 ‰ uncertainties. The exception is the 1995- 2005 period where the
mean residual difference of 5 ‰ is substantially higher than the mean reported
uncertainty of 2.5 ‰, reflecting the apparent larger scatter during this period as discussed
in section 3.5.3.

The one-sigma uncertainty on the smoothed curve and deseasonalized trend were
determined using a Monte Carlo technique. Each data point was perturbed by a random
normal error based on the reported uncertainty of that data point, such that the standard
deviation of all perturbations would equal the reported uncertainty to derive the one-
sigma and 95% confidence interval for the smooth curve.

### 3.7 Atmospheric Model Simulations

Simulations from the Numerical Atmospheric dispersion Modelling Environment
(NAME) III Lagrangian dispersion model (Jones et al., 2007) were used to interpret
seasonal variability in the dataset. The NAME model is run backwards in time to analyse
the history of the air traveling towards BHD and LAU over the preceding 4 days. For
each day of the simulation period, 10,000 particles were released during two time
windows in the afternoon; 13:00-14:00 and 15:00-16:00. NAME was driven by
meteorological output from the New Zealand Limited Area Model-12 (NZLAM-12), a
local configuration of the UK Met Office Unified Model (Davies et al., 2007.) NZLAM
has a horizontal resolution of ~12 km, with 70 vertical levels ranging from the earth's
surface to 80km. These simulations have been described in more detail by Steinkamp et
al. (2016). The average footprints presented here were computed by summing the
footprints for every day and release period in 2011-2013 and normalizing them such that
the domain integral equals one.





## 4. Results and Discussion

### 4.1. Variability in the Wellington record through time

The Wellington $\Delta^{14}CO_2$ record begins in December 1954, at a roughly "natural" pre-bomb $\Delta^{14}CO_2$ level of -20 ‰. From 1955, $\Delta^{14}CO_2$ increased rapidly, near doubling to 700 ‰ in 1965 at Wellington, due to the production of $^{14}C$ during atmospheric nuclear weapons tests. Nuclear tests in the early 1950s contributed to the rise, then a hiatus in testing in the late 1950s led to a plateau in Wellington $\Delta^{14}CO_2$ before a series of very large atmospheric tests in the early 1960s led to further increases (Rafter and Ferguson, 1959; Manning et al., 1990).

Most atmospheric nuclear weapons testing ceased in 1963, and the Wellington $\Delta^{14}CO_2$ record peaks in 1965 then begins to decline, at first rapidly at -30 ‰ yr$^{-1}$ in the 1970s and gradually slowing to -5 ‰ yr$^{-1}$ since 2005. The initial rapid decline has been attributed primarily to the uptake of the excess radiocarbon into the oceans, and to a lesser extent, uptake into the terrestrial biosphere (Stuiver and Quay 1981; Manning et al., 1990; Naegler et al 2006; Randerson et al., 2002). The short residence time of carbon in the biosphere means that from the 1980s, the terrestrial biosphere changed from a $^{14}C$ sink to a $^{14}C$ source as the bomb pulse was re-released (Randerson et al., 2002; Levin et al., 2010; Turnbull et al., 2009).

Natural cosmogenic production of $^{14}C$ damps the long-term decline, increasing $\Delta^{14}CO_2$ by 5 ‰ yr$^{-1}$; this may vary with the solar cycle, but there is no long-term trend in this component of the signal (Turnbull et al., 2009; Naegler et al., 2006). There is also a small positive contribution from the nuclear industry which emits $^{14}C$ to the atmosphere, and this has increased from zero in the 1950s to 0.5 – 1 ‰ yr$^{-1}$ in the last decade (Graven and Gruber, 2011; Turnbull et al., 2009; Levin et al., 2010).

The Suess Effect, the decrease in atmospheric $\Delta^{14}CO_2$ due to the addition of $^{14}C$-free fossil fuel $CO_2$ to the atmosphere (Suess 1955; Tans 1979; Levin et al., 2003), was first recognized in 1955 and has played a role throughout the record. Although the magnitude of fossil fuel $CO_2$ emissions has grown through time, when convolved with the declining atmospheric $\Delta^{14}CO_2$ history, the impact on $\Delta^{14}CO_2$ has stayed roughly constant at -10 ‰ yr$^{-1}$ since the 1970s (Levin et al., 2010; Randerson et al., 2002). Since the 1990s, the Suess Effect has been the dominant driver of the ongoing negative growth rate (Levin et al., 2010; Turnbull et al., 2009).

### 4.2. Seasonal variability in the Wellington record

We determine the changing seasonal cycle from smooth curve fits to five separate periods of the record (1954-1965, 1966-1979, 1980-1989, 1990-2004, 2005-2014). This subdivision is necessary to allow the seasonal cycle to vary through time since the ccgvu curve fitting routine assigns a single set of harmonics to the time period fitted (see section 3.6). The 1966-1979 period shows a strong seasonal cycle (figure 4) of about 30‰ amplitude, which is primarily attributed to seasonally varying stratosphere – troposphere exchange bringing bomb $^{14}C$ into the troposphere (Manning et al., 1990; Randerson et al.,





2002). Manning et al. (1990) were unable to simulate the correct phasing of the seasonal
cycle, apparently because their model distributed bomb $^{14}$C production throughout both
Northern and Southern stratosphere. In fact, the majority of the bomb $^{14}$C was produced
in the Northern Hemisphere stratosphere (Enting et al., 1982), and we show schematically
in figure 5 why this causes the opposite seasonal phase. Most transport across the
equator occurs in the troposphere, so that the Southern Hemisphere stratosphere would
have had a lower $\Delta^{14}CO_2$ than the Southern Hemisphere troposphere during the early
post-bomb period (figure 6). Since maximum cross-tropopause exchange occurs in the
spring (Olsen et al., 2003), this resulted in a minimum in $\Delta^{14}CO_2$ at Wellington in the
austral spring (August) when bomb $^{14}$C moved most rapidly into the stratosphere. The
seasonal cycle kept the same phase but gradually decreased in amplitude until the late
1970s, attributed to the declining disequilibrium between the stratosphere and
troposphere as the bomb $^{14}$C moved throughout the carbon reservoirs.
Between 1978 and 1980 the seasonal cycle weakened, and then reversed during the 1980s,
with a maximum in winter (June – August) and amplitude of about 5 ‰. This result is
comparable to that obtained by Manning et al. (1990) and Currie et al. (2011), who both
used a seasonal trend loess (STL) procedure to determine the seasonal cycle from the
same data. We hypothesize that as tropospheric $\Delta^{14}CO_2$ declined, and continued natural
production of $^{14}$C occurred in the stratosphere, the Southern Hemisphere stratosphere
eventually became enriched in $^{14}$C relative to the Southern Hemisphere troposphere, so
that consistent seasonally varying exchange processes resulted in a change in sign of
cross-tropopause $\Delta^{14}CO_2$ exchange in the late 1970s (figure 5). To the best of our
knowledge, no Southern Hemisphere stratosphere $\Delta^{14}CO_2$ measurements have been made
since the mid-1970s, so there is no direct evidence for this hypothesis.
The Wellington $\Delta^{14}CO_2$ seasonal cycle declined in the 1990s, and the larger variability in
the observations between 1995 and 2005 makes it difficult to discern a seasonal cycle
during that period. Since 2005, the more precise measurements allow us to detect a small
seasonal cycle with amplitude of about 2 ‰ (figure 4). Measurements from Cape Grim,
Australia from 1995-2010 show a similar magnitude seasonal cycle to that at Wellington
from 2005 - 2015, and a maximum in March – April that coincides with a seasonal
maximum in the Wellington record (Levin et al., 2010). However, Wellington $\Delta^{14}CO_2$
exhibits a second maximum in the austral spring (October) that is not apparent at Cape
Grim. Recent work has shown that during the winter, the Cape Grim station is
influenced by air coming off the Australian mainland including the city of Melbourne
(Ziehn et al., 2014), which would act to reduce $\Delta^{14}CO_2$ at Cape Grim relative to Southern
Ocean clean air. In contrast, the Baring Head location near Wellington is not
significantly influenced by urban regions in any season (figure 6). Air is typically from
the ocean, and the local geography means that the urban emission plume from Wellington
and its northern suburbs of Lower Hutt very rarely passes over Baring Head, and the
typically high wind speeds further reduce the influence of the local urban area (Stephens
et al., 2013). During the austral autumn, there is some land influence from the
Christchurch region in the South Island, but emissions from Christchurch are much
smaller than the Melbourne emissions influencing Cape Grim (State of Victoria fossil
fuel $CO_2$ emissions for 2013 were 23 MtC, Wellington and Christchurch each emitted 0.4


MtC of fossil fuel $CO_2$ in 2012/13 (AECOM, 2016; Australian Government, 2016; Boden
et al., 2012)). The observed Baring Head $\Delta^{14}CO_2$ maximum in spring in the recent part
of the record can be explained by the seasonal maximum in cross-tropopause exchange
bringing $^{14}$C-enriched air at this time of year (figure 5).

### 4.3. Comparison with other atmospheric $\Delta^{14}CO_2$ records
We compare the Wellington $\Delta^{14}CO_2$ record with several other $\Delta^{14}CO_2$ records that are
indicated in figure 1. First, we compare with measurements from Cape Grim, Australia
(CGO, 40.68°S, 144.68°E, 94 m asl). Cape Grim is at similar latitude to Wellington and
also frequently receives air from the Southern Ocean (Levin et al., 2010). Samples are
collected by a similar method to the Wellington record using NaOH absorption and are
measured by gas counting to ~2 ‰ precision. Next we compare with mid-latitude high-
altitude clean air sites in the Northern Hemisphere. The Vermunt, Austria (VER, 47.07°N,
9.57°E, 1800 m asl) record began in 1958, only a few years after the Wellington record
began, and in the 1980s the site was moved to Jungfraujoch, Switzerland (JFJ, 46.55°N,
7.98°E, 3450 m asl); these measurements are made in the same manner and by the same
laboratory as the Cape Grim record (Levin et al., 2013). We also consider the Niwot
Ridge, USA $\Delta^{14}CO_2$ record (NWR, 40.05°N, 105.59°W, 3523 m asl), which began in
2003 (Turnbull et al., 2007; Lehman et al., 2013). Niwot Ridge is also a mid-latitude
high-altitude site, but samples are collected as whole air in flasks and measured by AMS
in a similar manner to that described for the Wellington flask samples. Thus, we are
comparing two independent Southern Hemisphere records with two independent
Northern Hemisphere records, with the two hemispheres tied together by the common
measurement laboratory used for Cape Grim and Jungfraujoch. Results from all records
are compared in figure 7.

The Wellington and Cape Grim records are generally consistent with one another, with
the exception of the 1995-2005 period when the Wellington record is slightly higher,
apparently due to bias in the Wellington record (discussed in section 3.5.3.). Differences
between the sites are smaller than the measurement uncertainty for all other periods (table
2). This implies that the $\Delta^{14}CO_2$ signal is homogenous across Southern Hemisphere clean
air sites within the same latitude band, at least since the 1980s when the two records
overlap. Similarly, the high altitude, mid-latitude Northern Hemisphere sites are
consistent with one another, although there are some differences in seasonal cycles in
recent years (Turnbull et al., 2009).

The bomb spike is higher and earlier in the Northern Hemisphere records (figure 7),
consistent with the production of most bomb $^{14}$C in the Northern Hemisphere stratosphere
(figure 5). We determine a new estimate for the interhemispheric exchange time from the
difference in timing of the first maximum of the bomb peak in each hemisphere (July
1963 in the Northern Hemisphere, January 1965 in the Southern Hemisphere) as 1.4 years.
This is consistent with other more detailed interhemispheric exchange time estimates that
have been determined from long-term measurements of $SF_6$ of 1.3 to 1.4 years (Geller at
el., 1997; Patra et al., 2011).



536 Northern Hemisphere $\Delta^{14}CO_2$ remains higher than Southern Hemisphere $\Delta^{14}CO_2$ by
537 about 20 ‰ until 1972. Although most nuclear weapons testing ceased in 1963, a few
538 smaller tests continued in the late 1960s, contributing to this continued interhemispheric
539 offset (Enting, 1982). The interhemispheric gradient disappeared within about 1.5 years
540 after atmospheric testing essentially stopped in 1970. Except periods of noisy data from
541 Vermunt in the late 1970s and Wellington in 1995-2005, there are only small (<2 ‰)
542 interhemispheric gradients from 1972 until 2002 (figure 7, table 2).
543

544 From 2002, an interhemispheric gradient of 5-7 ‰ develops, with the Southern
545 Hemisphere sites higher than the Northern Hemisphere sites (table 2). We choose 1986 –
546 1990 and 2005 – 2013 as time periods to compare, to avoid the periods where the
547 Wellington record is noisy (1995 – 2005) and where we substituted flask measurements
548 from 1990 – 1993. In 1986 – 1990, there is less than 2 ‰ difference between Wellington
549 and either Cape Grim or Jungfraujoch. There is also no difference between the Cape
550 Grim and Jungfraujoch records during this time period. The Wellington and Cape Grim
551 records still agree within 2 ‰ after 2005, but both Jungfraujoch and Niwot Ridge diverge
552 from Wellington, by $4.8 \pm 2.7$ and $6.9 \pm 2.5$ ‰, respectively; they are not significantly
553 different from one another. This new interhemispheric gradient is robust, being
554 consistent amongst the sites measured by three different research groups each with their
555 own methods. It is not an artifact of interlaboratory offsets, since Cape Grim and
556 Jungfraujoch measurements are made by the same group using the same sampling and
557 measurement methods, and the Wellington and Niwot Ridge measurements (measured by
558 different techniques) agree well with the other sites at similar latitude (Cape Grim and
559 Jungfraujoch respectively). This developing gradient is also apparent in 2005 – 2007 in a
560 separate $\Delta^{14}CO_2$ sampling network (Graven et al., 2012), although that dataset extends
561 only to 2007. Graven et al. (2012) demonstrated that increasing (mostly Northern
562 Hemisphere) fossil fuel $CO_2$ emissions cannot explain this gradient, and instead, they
563 postulated that $^{14}C$ uptake into the Southern Ocean reduced over time.
564

565 The development of the interhemispheric $\Delta^{14}CO_2$ gradient coincides with an apparent
566 reorganization of Southern Ocean carbon exchange in the early 2000s (Landschützer et
567 al., 2015). The net Southern Ocean carbon sink is determined by the balance between
568 $CO_2$ uptake into surface waters, which are then subducted and sequester carbon, and
569 release of carbon to the atmosphere from upwelling of very old, carbon-rich deep waters.
570 $CO_2$ uptake into surface waters cannot change atmospheric $\Delta^{14}CO_2$, since the $\Delta^{14}C$
571 notation includes a mathematical correction for natural isotopic fractionation. In contrast,
572 the $^{14}C$ disequilibrium between old (and therefore $^{14}C$-poor), deep waters and the
573 atmosphere means that release of $CO_2$ from the Southern Ocean to the atmosphere
574 decreases atmospheric $\Delta^{14}CO_2$; the magnitude of that decrease depends on both the
575 carbon flux and the $^{14}C$ disequilibrium. Thus, since the 1980s, atmospheric $\Delta^{14}C$ has been
576 highly sensitive to Southern Ocean upwelling, the same mechanism that governs the
577 ocean $CO_2$ sink (Graven et al., 2012). Model simulations suggest that changes in
578 Southern Ocean ventilation may have played a key role in pre-industrial variations in the
579 latitudinal gradient of atmospheric $^{14}CO_2$ (Rodgers et al., 2011).





Several studies using both data and modeling suggests that the climate-induced increase
in westerly winds over the Southern Ocean increased upwelling of carbon-rich deep
waters and thus reduced the Southern Ocean $CO_2$ sink efficiency (Le Quéré et al., 2007;
Sitch et al., 2015). Yet, more recent evidence suggests a reinvigorated Southern Ocean
carbon sink since about 2002 (Munro et al., 2016; Landschützer et al., 2015). These
studies suggest that multiple factors contributed to the reinvigorated carbon sink, with
different controls in the different Southern Ocean regions; these data support a decreasing
upwelling of old, deep waters in recent years. Decreased upwelling would also cause a
relative increase in Southern Hemisphere $\Delta^{14}CO_2$ and thus drive the observed
interhemispheric $\Delta^{14}CO_2$ gradient, which appears at the same time as the apparent
reinvigoration of the carbon sink in the early 2000s.
Although the changing Southern Ocean carbon sink is the most likely explanation,
substantial underreporting of Northern Hemisphere fossil $CO_2$ emissions (e.g. Francey et
al., 2013) or changes in the land carbon sink (Sitch et al., 2015; Wang et al., 2013) could
also explain the new interhemispheric $\Delta^{14}CO_2$ gradient.

## 5. Conclusions

The 60 year-long Wellington $\Delta^{14}CO_2$ record has been revised and extended to 2014.
Most revisions were minor, but we particularly note that the earlier reported 1990-1993
measurements have been entirely replaced with new measurements. A second period
form 1995-2005 has poorer data quality than the rest of the record, and may also be
biased high by a few permil. These data have been revised substantially, and new
measurements have been added to this period, but we were unable to definitively identify
or correct for bias, so the data have been retained, albeit with caution. We further
validated the record by comparison with tree ring samples collected from the Baring
Head sampling location and from nearby Eastbourne, Wellington; both tree ring records
show excellent agreement with the original record, and indicate that there are no other
periods where the original measurements are problematic.
The Wellington $\Delta^{14}CO_2$ time series records the history of atmospheric nuclear weapons
testing and the subsequent decline of $\Delta^{14}CO_2$ as the bomb $^{14}C$ moved throughout the
carbon cycle, and $^{14}C$-free fossil fuel emissions further decreased $\Delta^{14}CO_2$. The timing of
the first appearance of the bomb-$^{14}C$ peak at Wellington is consistent with other recent
estimates of interhemispheric exchange time at 1.4 years.
The seasonal cycle at Wellington evolves through the record, apparently dominated by
the seasonality of cross-tropopause transport, which drives a changing seasonal cycle
through time. In the early post-bomb period, the Southern Hemisphere troposphere was
enriched in $^{14}C$ relative to the Southern Hemisphere stratosphere so that the seasonal
minimum occurred at Wellington when cross-tropopause transport is at a maximum. The
seasonal cycle reversed once the bomb perturbation reduced and continuing natural
cosmogenic production meant that the Southern Hemisphere stratosphere was once again
enriched in $^{14}C$ relative to the troposphere. In recent years, the seasonal cycle has
amplitude of only 2 ‰, with a maximum in the austral spring. Cape Grim exhibits a
similar seasonal cycle magnitude, but appears to be slightly influenced by a





terrestrial/anthropogenic signal during the austral winter that is not apparent at
Wellington.
During the 1980s and 1990s, $\Delta^{14}CO_2$ was similar at mid-latitude clean air sites in both
hemispheres, but since the early 2000s, the Northern Hemisphere $\Delta^{14}CO_2$ has dropped
below the Southern Hemisphere by 5-7 ‰.  This is most likely due to a change in
Southern Ocean dynamics reducing upwelling of old, $^{14}$C-poor deep waters, which is
consistent with recent evidence for an increasing Southern Ocean carbon sink.  This
result implies that ongoing and expanded Southern Hemisphere $\Delta^{14}CO_2$ observations and
modelling can provide a fundamental constraint on our understanding of Southern Ocean
dynamics and exchange processes.

## 6. Acknowledgements

A 60 year-long record takes more than a handful of authors to produce.  This work was
possible only because of the amazing foresight and scientific understanding of Athol
Rafter and Gordon Fergusson, who began this record in the 1950s.  Their work was
continued over the years by a number of people, including Hugh Melhuish, Martin
Manning, Dave Lowe, Rodger Sparks, Charlie McGill, Max Burr and Graeme Lyon.
This work was funded by the Government of New Zealand as GNS Science Global
Change Through Time core funding and NIWA Greenhouse Gases, Emissions, and
Carbon Cycle Science Programme core funding. The author(s) wish to acknowledge the
contribution of New Zealand eScience Infrastructure (NeSI) to the results of this research.
New Zealand's national compute and analytics services and team are supported by the
NeSI and funded jointly by NeSI's collaborator institutions and through the Ministry of
Business, Innovation and Employment (http://www.nesi.org.nz).



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

Institute of Geological and Nuclear Sciences, New Zealand. Radiocarbon 38, 133-
874      134.





## 8. Tables

| Date Range | NZ/NZA | Site | collection method | Measurement method |
|---|---|---|---|---|
| 1954-1986 | 0-7500 | MAK | tray | GC |
| 1987-1994 | 7500-8400 | BHD | tray | GC |
| 1995-2004 | 8400-30000 | BHD | bottle | ENTandem $^{13}$C $^{14}$C |
| 2005-2009 | 30000-34000 | BHD | bottle | ENTandem $^{12}$C $^{13}$C $^{14}$C |
| 2010-2011 | 34000-50000 | BHD | bottle | XCAMS |
| 2012-present | 50000- | BHD | bottle | XCAMS/RG20 |

**Table 1.** Wellington $^{14}CO_2$ measurement methods through time.

| Site difference | Time period | $\Delta^{14}CO_2$ difference |
|---|---|---|
| WLG-CGO | 1986-1990 | $1.8 \pm 2.5$ |
| WLG-CGO | 2005-2013 | $1.3 \pm 3.4$ |
| WLG-JFJ | 1986-1990 | $0.8 \pm 3.9$ |
| WLG-JFJ | 2005-2013 | $4.8 \pm 2.7$ |
| WLG-NWR | 2005-2013 | $6.9 \pm 2.5$ |

**Table 2.** $\Delta^{14}CO_2$ gradients between sites, determined as the mean of the monthly differences for each time period. Errors are the standard deviation of the monthly differences.




## 9. Figures

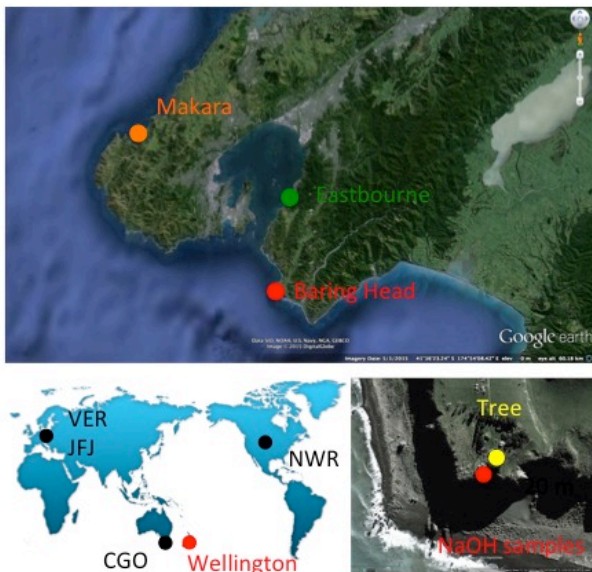

**Figure 1.** Sampling locations. Top: Makara (1954-1986) and Baring Head (1987 –
present) air sampling sites and the location of the Eastbourne tree samples.  Bottom left:
world location showing Wellington and other sampling sites discussed in the text.
Bottom right: close up of the Baring Head site showing the relative positions of the air
(NaOH) and tree sampling locations.





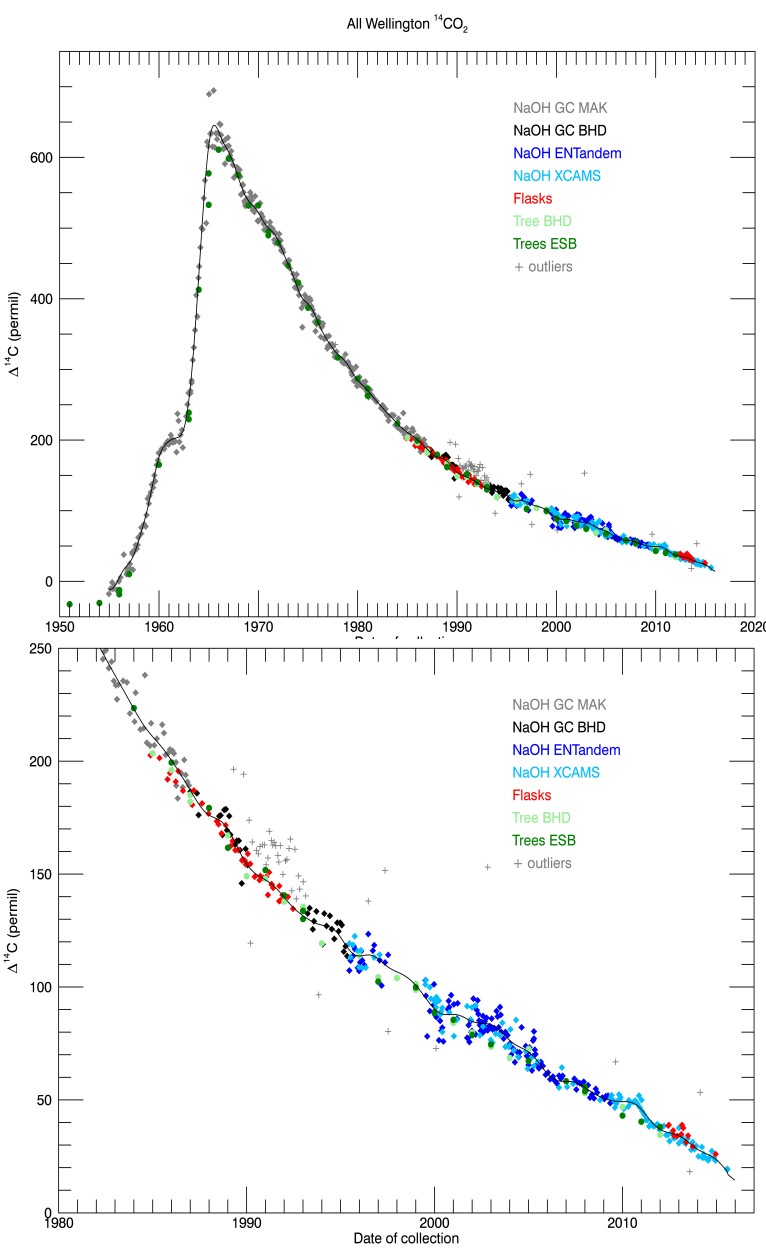

**Figure 2.** Wellington $^{14}CO_2$ record showing all collection and measurement methods.
Tree rings (green) and outliers (grey pluses) are excluded from the reported final dataset.
Black line is the smooth curve fit to the final dataset.



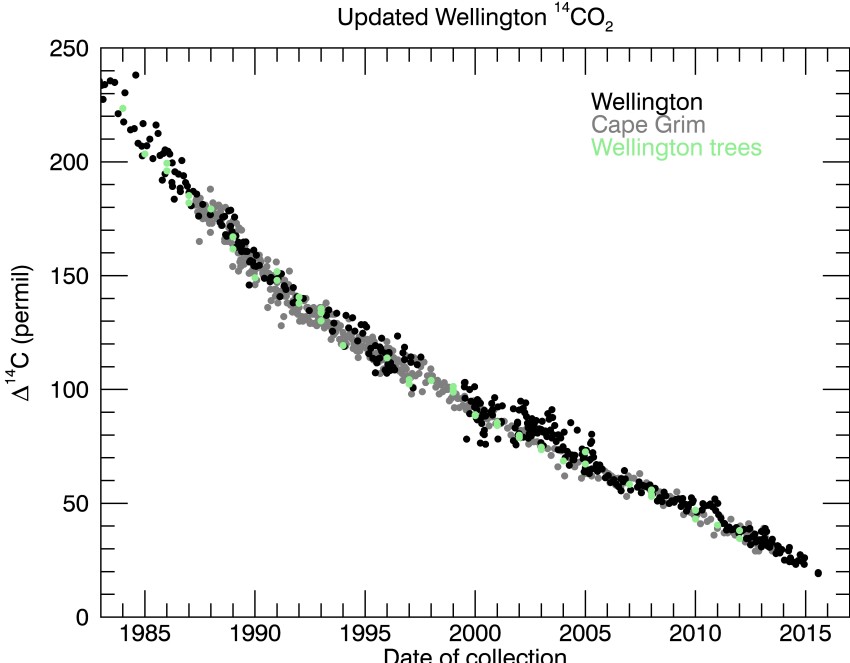

Figure 3.  Comparison of the final Wellington and Cape Grim (Levin et al., 2010) $\Delta^{14}CO_2$
records.  Wellington tree ring measurements are also shown.



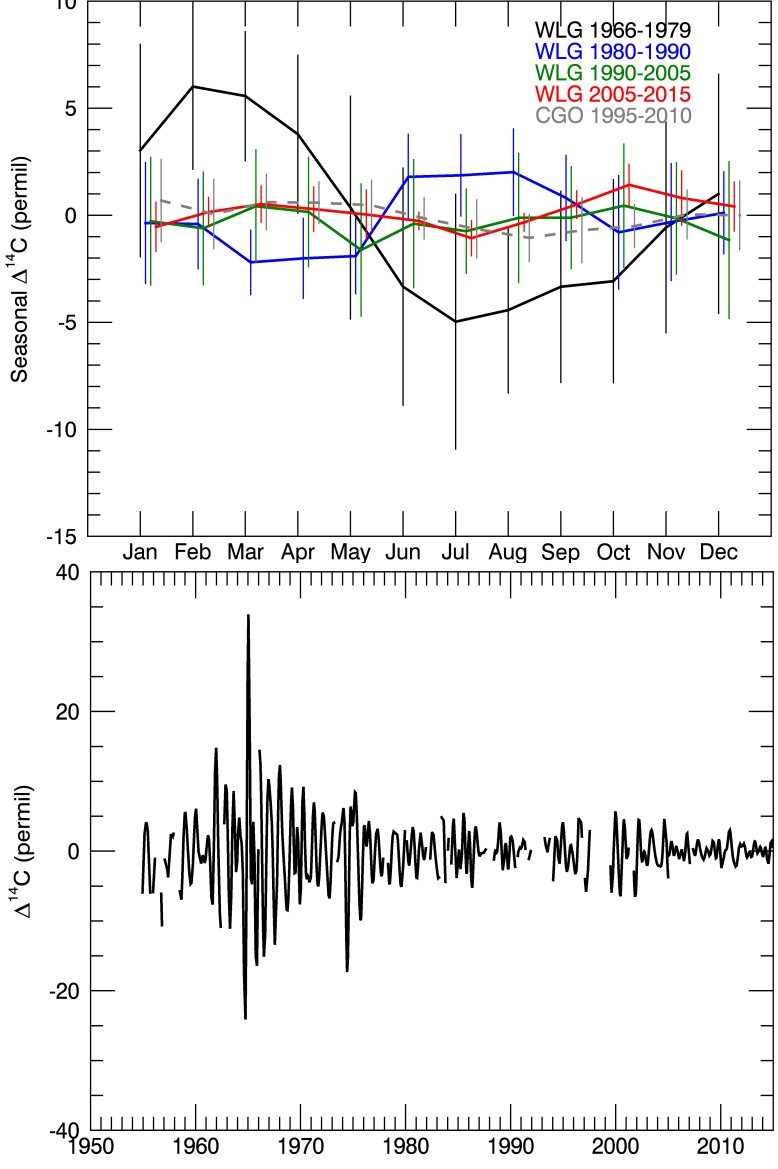

Figure 4.  Detrended seasonal cycle in the Wellington $\Delta^{14}CO_2$ record.  Top: WLG
monthly detrended seasonal cycle averaged over four time periods as described in the text
and the CGO (Levin et al., 2010) detrended seasonal cycle.  Error bars are the standard
deviation of all years averaged.  Points for each time period are slightly offset for clarity.
Bottom: full seasonal cycle record determined separately for each time period shown in
the top panel plus 1954-1965.





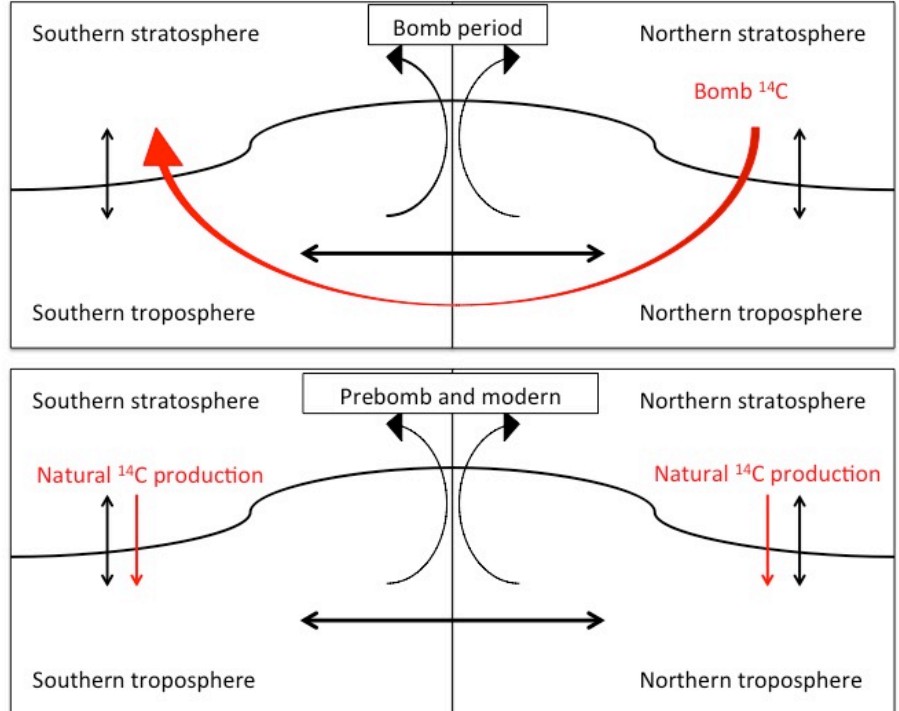

Figure 5.  Schematic of stratosphere – troposphere circulation.  Black lines show
directions of mixing, and the red lines show the movement of $^{14}$C.  Top panel shows
movement during the bomb period when bomb $^{14}$C produced in the Northern stratosphere
dominated over natural $^{14}$C production and atmospheric transport quickly enriched the
Southern troposphere relative to the Southern stratosphere.  The bottom panel shows the
pre- and post-bomb periods when natural $^{14}$C production in both Northern and Southern
stratospheres results in the Southern stratosphere being enriched relative to the underlying
troposphere.





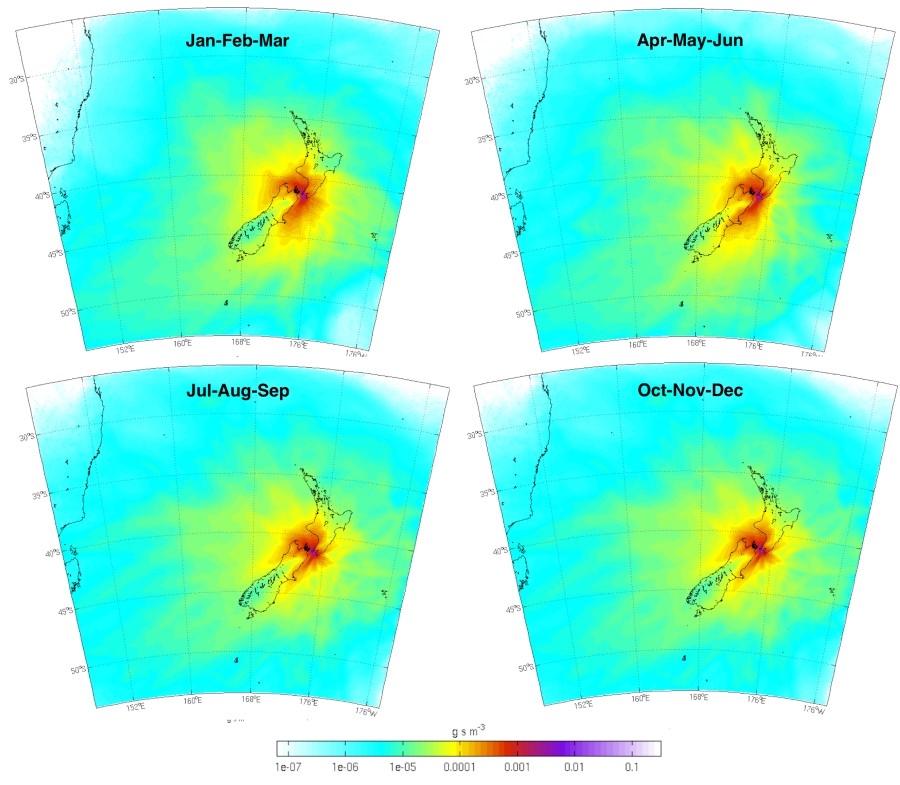

Figure 6. Mean footprints for the BHD site for each three-month period, averaged over
the years 2011 – 2013.  Footprints were determined using the NAME III atmospheric
dispersion model forced with meteorology from the NZLAM weather prediction model.

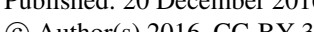




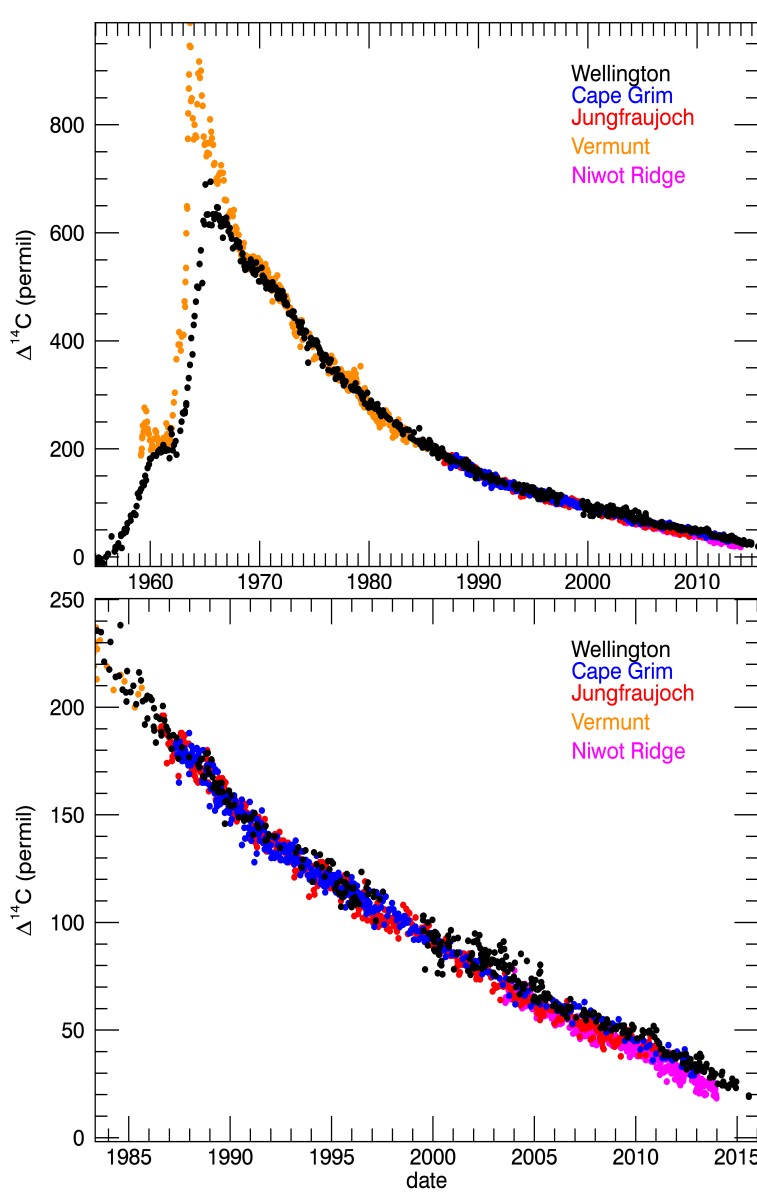

Figure 7.  Comparison of Wellington and other atmospheric $\Delta^{14}CO_2$ records (Levin et al.,
2010; Turnbull et al., 2007; Lehman et al., 2013).