# Peer review of "Sixty years of radiocarbon dioxide measurements at Wellington, New Zealand 1954 – 2014"

_Atmospheric Chemistry and Physics, 2016_

## Referee Comment (RC1) · S. Hammer (Referee) · 12 Feb 2017

Review on "Sixty years of radiocarbon dioxide measurements at Wellington, New Zealand 1954 – 2014" by Turnbull et al. 2016

Turnbull et al. present a thorough revisit of the entire Wellington atmospheric 14CO2 record. They re-measured archived samples and include new information from tree samples to better investigate known "noisy" periods of original record. Conceivable flagging criteria are formulated and the Wellington record is compared to independent data sets. Therefore this manuscript is of upmost scientific interest to the radiocarbon community and I definitely recommend publication in ACP. In addition to the data review the authors revisit and extend the key findings that the Wellington 14CO2 record

provides. For some of the conclusions drawn from the data I would like to ask the authors to reinforce their arguments to overcome my minor concerns.

General comments to the authors:

P5 l212ff 14C measurements:

Have you investigated if the use of IRMS-13C in the early AMS measurements introduces a bias? Such a potential bias could originate e.g. from a machine immanent fractionation. I assume you have IRMS-13C measurements also for the post-2005 samples. Did you compare the effect of offline and online 13C measurements for the D14C normalization directly? Such an investigation will also quantify the contribution to the scatter which is due to offline 13C analysis in the earlier AMS results.

P8 l338 Smooth curve fit:

Fitting section by section may introduce problems at each overlap of the sections. Wouldn't it be better to use a fit routine which can deal with a changing phase? Pickers et al. mention that STL per se does not require gap filling, only the current implementation of STL does. Pickers et al. also investigate HPspline which would allow for a change in phase. Why didn't you chose this fitting algorithm? When you investigate the phase change in the 14CO2 signal, you find that the seasonal cycle weakens between 1978 and 1980, and then reverses. Could it be that this timing is related to the change in the fitting sections (1966-1979 and 1980 to1989). The described method for overlap and interpolation between different fits favors the weakening of the seasonal cycle at the section borders if both sections are out of phase. I wonder if you would find the same timing for the phase change if you chose different fitting sections. . .

P11 l450ff Hypothesis of reversed seasonal cycles in the early post-bomb era:

The hypothesis behind the changing phase in the seasonal cycle should be backed up by a small (box-) model exercise. This model should include the seasonal cycles of the STE (in NH and SH) and the CEE (cross equator exchange) in the troposphere

and the stratosphere. The Mount Pinatubo eruption is a well-studied phenomenon when it comes to stratospheric transport. see e.g. Aquila et al. 2012. They find middle-stratospheric meridional pathways with mixing times of less than a year. The major stratospheric bomb-peak lasted for about 4-5 years (see HASL data compiled in Naegler et al 2006). Can you show in a (box-) model that with those boundary conditions your hypothesis is valid? Aquila, Valentina, et al. "Dispersion of the volcanic sulfate cloud from a Mount Pinatubo–like eruption." Journal of Geophysical Research: Atmospheres 117.D6 (2012).

P11 l471ff Interpretation of the seasonal cycles since 2005:

I have a couple of questions and comments to the comparison of the Wellington and Cape Grim seasonal cycles: - The comparison to the Cape Grim seasonal cycle is problematic since both mean cycles do not average the same time period. Figure 4b shows that there are obvious large inter-annual variations in the amplitude (phasing?) of the seasonal cycle. - What is the origin of the double maxima in the BHD cycle? - Is the Melbourne influence at Cape Grim detectable in $CO_2$ or CO? - Fig 6 does not convince me that BHD is not influenced by anthropogenic emissions. Wellington is in the middle of the "red" area. When reading Pickers et al. they mention that in their data example of the BHD $CO_2$ data they had to gap fil 10% of the data since they deviated from baseline conditions. . . . To me this indicates some anthropogenic influence at BHD as well. - Sure, Melbourne emits 50 times more ff$CO_2$ than Wellington, however the distance between Melbourne and Cape Grim is 340km, whereas it is around 10km between Wellington and the BHD. . . - If STE is the driving mechanism for the seasonal cycle for the periods 1966 to 1979 and 1980 to 1990, how come that the seasonal cycle post 2005, which is also explained via the STE, is not in phase with the earlier once. . .

Specific comments:

p.2 l.40 Please state the years when the measurements in Norway and Austria started
p2. l.44 The term "exchanges" is a bit too general, consider oxidized or something

more specific. p.2 l.45: Production -> Natural production p2. L 47: perturbations to .14CO2 -> perturbations to natural .14CO2 levels p2. L62: Add year to Lopez et al., and add also early attempts of ffCO2 emission estimates like e.g: Meijer, H. A. J., et al. "Isotopic characterisation of anthropogenic CO 2 emissions using isotopic and radiocarbon analysis." Physics and Chemistry of the Earth 21.5 (1996): 483-487. Gamnitzer, U., U. Karstens, B. Kromer, R. E. M. Neubert, H. A. J. Meijer, H. Schroeder, and I. Levin (2006), Carbon monoxide: A quantitative tracer for fossil fuel CO2? J. Geophys. Res., 111, D22302 p.2 l77: add citations to the last part of this paragraph p4 l128: what do you intend with the term "nominally CO2-free"? Did you process blank NaOH solutions? How much CO2 is in a blank NaOH solution? What is the 14C activity of this blank? p4.l131: "large tray" can you state the surface area of that tray? P4.139: Please add the statement about fractionation (supplement S3.l90-92) to the main text. P5 l189 "one" sd? In Fig S2 and the text you state 2 sd? P6 l259 please include a reference to Fig.2 in this subsection P8 l316 I don't see the 2005 EN Tandem improvement mentioned in Zondervan et al 2015.... Maybe I overlooked it? P8 l336 Do the measurements from this period carry a special flag (e.g noisy) in the dataset? Reading the supplement I found that you are already doing this. Maybe make a short note in the main text. P8 l353 how does ccgvu handle data gaps and inconsistent sampling frequencies? Since the paper is (at least for me) not freely available it is worth mentioning this shortly in the supplement. P9 l362 what is the unit of the cutoff criteria in the frequency domain? P9 l363 is the 2 year overlap a good idea? In terms of transition yes, but don't you have now the influence of end-effects in 4 years? P9 l368 "mean residual difference" do you mean RMS of the residuals P9 l379 state the "n" of the MC P9 l382 where are the 95% conf intervals given? In the data set I see only one uncertainty column, please specify in the data-set if this is the 1 sigma error or the 95% conf interval. P9 l384ff the model simulation are not convincingly not used in the paper. See general comments. Consider skipping the subsection 3.7 and Fig 6. P9 l388 LAU ?? P10 l403ff include ref to fig. 2 P10 l442 30 per mil amplitude for the period 1966-1979? I only see such an amplitude once? A mean amplitude of ca. 7 per mil

seem more realistic. P11 l456 fig 6 -> fig 5?? P11 l459 "Between 1978 and 1980 the seasonal cycle weakened". This is not really seen in fig 4b. Unfortunately 1978 to 1980 is a boundary of the fitting sections... since the seasonal cycles for the two sections are opposed and the overlap is linearly interpolate between fits... a weakening can also come from the applied method. P11 l460 5 per mil amplitude? Maybe two times in this period... 3 per mil on average P11 l467 fig 5 -> fig 4 P12 l 494 fig5 -> fig 4? P12 l497 "records that are indicated in figure 1" -> "records where the sampling locations are indicated in figure 1" P13 l563 Model results from Levin et al. 2010 already suggest the development of a interhemispheric gradient in the same magnitude for the same time... without changing the southern ocean... although they admit that they are not matching the data... P21 Table1: include sample no. to NZ/NZA, replace GC with gas counting, change "measurement methods" to "measurement and sampling methods" Table2: provide the unit to the 14C differences P22 Figure1: provide scales to the google earth pictures, indicate urban areas in the upper map. P23 consider vertical grid lines to illustrate the different periods used in the paper. Consider indicating graphs with a) and b) x-label of graph a) is cropped... p25 Consider indicating graphs with a) and b) in a) use the same periods as in the text. b) consider vertical grid lines to illustrate the different periods p27 Motivate the plot better. Not really used in the paper. Explain the unit. P28 Consider indicating graphs with a) and b) Consider usage of open symbols. Especially after 2000 it would be good to see all data.

Supplement:

S2.l74 state the surface area of the pyrex tray S4 l147 extraction follows -> extraction from 1995 onward follows S5 l217ff in total after flagging you have 427 targets, if you split them between the machines you have 397 and 102 …. To me this does not add up? What am I missing? S6. L262 Please state the main offset for the QC datasets between the two AMS machines. S9 l394 What is RLIMS? S12.l457 Indicate the figure S1 with a) and b). I assume a) is Eastbourne and b) is Baring Head? Correct? S12.l468 Since you cannot decide between "red" or "green" for the Baring Head tree, how can

you than state the excellent agreement? Is it excellent for both red and green? Please include a link to the t-test or the mean difference to reinforce this statement. S13 l471 Define "NIK". Why is there only one comparison for NIK and 4 comparisons for BHD? S13 l473 please specify the t-test: I assume you use a dependent t-test for paired samples? Since the applied formulas are easy it might be clearer if you just explicitly state them. S13 l481 what is the mean difference if you use the one year shifted BHD tree (red points in fig S1)?

Technical comments:

In the text please use a consistent ordering (e.g. temporally ascending) when citing multiple papers.

Please also note the supplement to this comment:
http://www.atmos-chem-phys-discuss.net/acp-2016-1110/acp-2016-1110-RC1-supplement.pdf

---

## Referee Comment (RC2) · Anonymous Referee #2 · 13 Feb 2017

The atmospheric radiocarbon measurements conducted at Wellington are a very important record and the authors' efforts to maintain and evaluate the observations are valuable to the community.

However, there are some major revisions needed before publication of this manuscript. Much of the paper is used on re-reporting trends and gradients that have already been shown in other work. The authors also make unsupported claims about the mechanisms driving the interhemispheric gradient and seasonal cycles of D14C.

The paper postulates a sensitivity to Southern Ocean air-sea exchanges that is misleading and unsupported. It gives the impression that the Southern Ocean only began influencing the interhemispheric D14C gradient in 2002, whereas the Southern Ocean

has always been a primary influence on the interhemispheric D14C gradient, via gross, not net, carbon exchange. Levin et al. 2010 and Randerson 2002 clearly show that the observed trend in the interhemispheric D14C gradient is consistent with a long-term change in the oceanic influence, dominated by the long-term decrease in atmospheric D14C and the change in D14C disequilibrium over the Southern Ocean, which is further supported by the Graven 2012 papers. A change in upwelling is interesting to consider as a secondary effect, but the authors do not include quantitative models or estimates of how large the effect could be, nor any specifics on how it influences D14C. Furthermore, the Wellington data from 1995-2005 are shown to have serious issues, which would complicate identification of a signal originating in the early 2000s. And there is no discussion about the period in the 1990s when upwelling was increasing.

The authors similarly make statements about the influences on the seasonal cycle of D14C at Wellington that aren't well-supported.

The paper should be shortened to minimize the re-reporting of previous observations, reduce repetition, clarify the long-term trend in the Southern Ocean influence on the interhemispheric D14C gradient, and remove unsupported statements. As the main contribution is to revise the Wellington data, i.e. no new modeling or other evidence is given to help interpret the data, the paper might be better suited to a journal like Radiocarbon or Atmospheric Measurement Techniques.

Specific Comments.

Section 3.5.3 appears to show major problems in the measurements for the 1995-2005 period, with large scatter and a high bias. I don't agree that the questionable data should be retained, as the authors have done - "in the absence of better data, we retain both the original and remeasured NaOH sample results in the full record." This conflicts with the aim of the paper to evaluate and refine the previously reported measurements and, presumably, to prevent the interpretation of measurement problems as real atmospheric variability.

The code WLG is already used by NOAA for Mt Waliguan, China – perhaps another code would be better.

L15 Earliest direct atmospheric

L98 Revisiting key findings can be placed in the introduction for brevity.

L104-108 Unsupported. See above comment.

L234 Please quote a value for precision

L306 Why would this result in higher D14CO2?

L378 More detail needed. Where is this used?

L384 How do 4-day back trajectories address the seasonal cycle? The panels in the figure all look the same. This is not very useful. A panel should be shown with the differences if there is a difference to highlight.

L413 Since 2005 or earlier?

Section 4.1 seems out of place and repetitive. Should move to introduction and focus on new results here.

L435 Turnbull 2009 only includes simulations from the 2000s, so they do not show the Suess Effect became the dominant driver in the 1990s.

L454 Do you mean when mixing with lower-D14C air from the stratosphere was the strongest? Are there Southern Hemisphere stratospheric observations from the bomb period supporting the idea that tropospheric D14C was higher than stratospheric D14C? Are you saying that tropospheric D14C was higher than stratospheric D14C in the Southern Hemisphere until the late 1970s? Bomb 14C would have also entered the SH stratosphere through the tropical tropopause, while at the same time tropospheric D14C was declining, so this seems unlikely. Note Northern Hemisphere sites also showed minima in spring in the early bomb period. Levin 2010 simulate recent

seasonal influences on D14C and should be cited here. Oceanic influences on the seasonal cycle should also be mentioned.

L468 See Brenninkmeijer, C. A. M., Lowe, D. C., Manning, M. R., Sparks, R. J., & van Velthoven, P. F. J. (1995). The 13C, 14C, and 18O isotopic composition of CO, CH4, and CO2 in the higher southern latitudes lower stratosphere. Journal of Geophysical Research: Atmospheres, 100(D12), 26163-26172. doi:10.1029/95JD02528

L494 This is the time of maximum in the NH so this phasing is unexpected. Is there an explanation for the double-peaked shape of the cycle? This section relies on dismissing the Cape Grim data, which is not entirely convincing. Are other Southern Hemisphere observations relevant here?

L517 It would be useful to include a plot of the difference between the Wellington and Cape Grim data.

L521 Delete the word signal. Is it possible to say something more quantitative than "homogeneous"?

L527 What is the basis for the new estimate of the interhemispheric exchange time? How was this calculated? Without any supporting information this paragraph should be deleted.

L544 Need to cite Levin 2010, and Graven 2012

L561 Also shown in Randerson 2002 and Levin 2010

L565 This paragraph is misleading. See main comment above.

L575 This is the gross carbon flux not the net carbon flux. Atmospheric D14C has been highly sensitive to Southern Ocean upwelling not only since the 1980s but since the preindustrial period and throughout the bomb peak period – see Randerson 2002 and Levin 2010

L593 "Although the changing Southern Ocean carbon sink is the most likely explanation," Atmospheric D14C is not directly affected by the Southern Ocean carbon sink. What is the justification for this statement? See main comment above.

---

## Referee Comment (RC3) · J. Miller (Referee) · 1 Mar 2017

General comments.

This paper documents and analyzes the longest atmospheric radiocarbon time series from a single site. Obtained near Wellington, New Zealand starting in 1954 and continuing to the present, these data represent a signature time series of carbon cycle science. The authors document the revision and evaluation of the data, which should lead to a significant improvement in its scientific utility. The seasonal cycle and trend are analyzed convincingly, although too much attention is paid to the hypothesis that an increased Southern Ocean $CO_2$ sink can explain the changing $\Delta 14C$ atmospheric north-south gradient. While it's true that the change in the north-south 14C gradient

supports this idea, there is no new analysis of the time series to bolster it. One additional point is that it would be good to provide the internet location of the data in addition to the static spreadsheet provided. Presumably the ftp site would contain the data set of record including the latest data, flags, and corrections. Nonetheless, this is a strong paper that is entirely appropriate for ACP; it should be published after a few modifications.

Below, I list some edits and comments by line number.

Specific comments.

L21,22. While Cape Grim air samples may contain anthropogenic signals in winter, air samples have often been collected during times when the wind is not coming from the north.

L44. 'exchanges' is a bit vague. Why not spell it out to say that 14C reacts immediately with O2 to form 14CO, which is subsequently oxidized to 14CO2

L68-70. This is redundant with text around L44.

L75. Perhaps strike 'now', and add 'in the two decades following the atm. test ban treaty' at the end of the sentence.

L77. I don't agree that the additions of fossil fuels became the dominant factor influencing the 14CO2 trend. If fossil fuel CO2 additions are 'dominant' I would think of them being an order of magnitude or so larger than other processes. Presently (and more or less in the 1990s), fossil fuel combustion alone would reduce the atmospheric $\Delta$14C by $\sim$ 10 per mil/yr; cosmogenic production would increase it by 5 per mil/yr; the land-atmosphere and ocean-atmosphere disequilibrium fluxes would be roughly +4 and -4 per mil/yr. It might be reasonable to try and calculate a point at which the negative trend in atmospheric $\Delta$14C was driven more by fossil fuel emissions than by absorption of bomb 14C atoms into the biosphere and oceans. But this would not equate to 'dominant' in my opinion. L80. Change 'especial' to 'special'

L129. Use 'M' (molar) or 'mol/L'

L158. 'Faithfully' record $\Delta$14C, but not the 14C content, which is offset by $\sim$ 34 per mil.

L210. Was testing done do see if the samples could be stored for up to three years before analysis without introducing artifacts.

L216-218. Could using an offline $\delta$13C value produce bias or just add noise? Any tests to examine this?

L227. Considering that the multi-target averaging resulted in differences of up to 5 per mil, I think that this deserves a detailed explanation, at the very least in the supplement.

L243. S+P's $\Delta$ is the same as the presently used $\Delta$14C; their $\Delta$14C is defined differently.

L255. How was the weighting done? Inverse square of the measurement precision?

L280. Wondering if 'excursion' is the best word here. Anomaly?

L283. As mentioned in comments on L22, Cape Grim sampling can be 'tuned' just for a clean air sector. If the issue is integrated sampling, then I would say that.

L284. Change 'terrestrial' to 'mainland'?

L303-304. 'preparation was conducted' to 'was prepared'.

L313. 'or thereafter' to 'and thereafter'

L325. I don't see the reduction of scatter shown in any plot. It would be useful to show how the reprocessing improved the noise.

L351. Change 'ccgvu' to 'ccgcrv' which is the actual name of the curve fitting code.

L362. Insert 'day' after 80. Good that this important detail was included.

L395. Add a sentence explaining what a footprint is.

L403. I think 'roughly "natural"' can be deleted; natural is ambiguous. Maybe 'roughly pre-industrial'?

L421-422. By 'long-term' to you mean decline since the 1960s? For many in the radiocarbon world, that wouldn't be very long, so maybe define the time period more explicitly. Also, insert 'known' prior to 'long-term trend in...'

L434. As mentioned earlier, I don't think 'dominant' can be justified.

L469. I'm wondering about the value of an untestable hypothesis. What you say sounds plausible, but maybe refer to it as speculation?

L507. Should Levin et al reference by 2010? 2013 paper appears to deal with Europe.

L527-534. I would like to see the math of how this was calculated, at least in the supplement. Also, one important factor is to know the state of ENSO during the 1963-1965 period, because La Nina, for example, can significantly increase inter-hemispheric exchange. Finally, the SF6 derived value is based purely on surface data, whereas the $\Delta14C$ method has a significant upper atmosphere component. It would be good to comment on how the estimates might differ.

L544 – 596. I felt that the text at the end of the Results and Discussion section focusing on the interhemispheric gradient and the Southern Ocean was a bit out of place. The Wellington $\Delta14C$ data confirm the gradient observed earlier and extend it in time. However, at present, the two paragraphs (starting at line 565) sound more like a review of the Southern Ocean uptake hypothesis, because there doesn't appear to be any new analysis. If it's not possible to add any new analysis using the Wellington data, I think it would be better to be very concise, essentially saying something like 'our data suggest the S.O sink continues to explain... Numerous recent studies using methods x, y and z further support... Our data set will be a powerful constraint to understanding the evolution of the gradient in a quantitative model framework...'

L571. Change 'natural' to 'mass-dependent'?

L650. Perhaps acknowledge Scott Lehman and Ingeborg Levin for providing unpublished data.

Table 2. WLG is already taken as a site code (for Mt. Waliguan Observatory, China), at least with respect to the WMO GAW program. Wouldn't MAK and BHD work here?

Figure 2. Can you distinguish the symbols and/or colors for the two versions of the EN-Tandem: i.e. 12,13,14 vs. 13,14, since the results seemed to be significantly different.

[Figure]

---

## Author Comment (AC1) · 12 Aug 2017

Dear Editor,

We thank the three reviewers for their thoughtful reviews and respond to each point individually below, as well as making changes in the revised manuscript. First, we make three general points, and then respond to each reviewer comment (reviewer comments in bold, our responses in plain text).

1. We asked for additional time to revise the paper to address a key suggestion of reviewer 1, that is, to use a box model to help explain the observed seasonal cycle. Ultimately, we have not included the box model analysis in the paper for the following reasons. As several previous authors (Randerson et al., 2002; Levin et al., 2010) have pointed out, it is difficult to reproduce the $^{14}$C bomb spike, seasonal cycle and rate of decline with a simple 4 box model (Northern and Southern Hemispheres, each divided into troposphere and stratosphere). They were right, and we were unable to match the bomb spike peak, timing or interhemispheric offset unless we adjust the transport and flux terms to such an extent that we do not believe it is justifiable to interpret the results in a meaningful way. (For example, to match the maximum bomb peak amplitude in the Northern troposphere, we needed either an unrealistically fast stratosphere-troposphere exchange rate of less than one year or to place 20% of the bomb $^{14}$C in the Northern troposphere (rather than stratosphere)). We considered building a more elaborate box model, but concluded that our existing capabilities make it more realistic for us to focus on including $^{14}$C a higher resolution global atmospheric transport model for a future publication. Thus, we have substantially revised the discussion around the seasonal cycle to address the reviewer comments and remove sections that are speculative, and we have not included box modelling.

2. Reviewer 2 suggests that there is not sufficient new information or interpretation in the paper to warrant publication in ACP. We respectfully disagree. First, both reviewers 1 and 3 recommend publication with revisions. Second, the Wellington $^{14}CO_2$ record is the longest direct atmospheric record of any trace gas or isotope anywhere in the world, and is the only long-term Southern Hemisphere $^{14}CO_2$ record. It has been used widely and will no doubt continue to be used widely (previous reports on the Wellington record have been directly cited 138 times (Currie et al 2011, Manning et al 1990) and the dataset is the main Southern Hemisphere record used in compiled $^{14}$C global records that have been cited more than 500 times (e.g. Hua and Barbetti 2013, Hua et al 2004)). As such, we believe this continues to be an important record that should be widely discoverable, and ACP is a suitable place for it.

3. Reviewer 2 also asks for a shortening of interpretation that is repeated from previous publications. We understand the reviewer's point of view, but we believe that when reporting on a long record, it is frustrating to the reader to have to refer to previous publications to find interpretation of the long record. We have altered the text to make clear where interpretation has been reported elsewhere and where it is new.

**Reviewer 1 Samuel Hammer**
**Turnbull et al. present a thorough revisit of the entire Wellington atmospheric 14CO2 record. They re-measured archived samples and include new information from tree samples to better investigate known "noisy" periods of original record. Conceivable flagging criteria are formulated and the Wellington record is compared to independent**

**data sets. Therefore, this manuscript is of upmost scientific interest to the radiocarbon community and I definitely recommend publication in ACP.**
**In addition to the data review the authors revisit and extend the key findings that the Wellington 14CO2 record provides. For some of the conclusions drawn from the data I would like to ask the authors to reinforce their arguments to overcome my minor concerns.**

**General comments to the authors:**
**14C measurements:**
**Have you investigated if the use of IRMS-13C in the early AMS measurements introduces a bias? Such a potential bias could originate e.g. from a machine immanent fractionation. I assume you have IRMS-13C measurements also for the post-2005 samples. Did you compare the effect of offline and online 13C measurements for the D14C normalization directly? Such an investigation will also quantify the contribution to the scatter which is due to offline 13C analysis in the earlier AMS results.**

Yes, of course. We believe that indeed the use of IRMS-13C measurements in the 1995-2005 AMS analyses is the reason for the variability. The very clear reduction in noise from 2005 when online AMS 13C analysis was added is very convincing evidence, and there is ample evidence from many AMS labs that this is likely the explanation. This is discussed in two places (sections 3.3 and 3.5.3). We have expanded the text and pointed the reader to the other section in the discussion.

**Smooth curve fit:**
**Fitting section by section may introduce problems at each overlap of the sections. Wouldn't it be better to use a fit routine which can deal with a changing phase? Pickers et al. mention that STL per se does not require gap filling, only the current implementation of STL does. Pickers et al. also investigate HPspline which would allow for a change in phase. Why didn't you chose this fitting algorithm?**
We did consider using other algorithms, particularly STL, since this was used in previous analysis of the Wellington $^{14}CO_2$ record. Pickers et al showed that of the three, HPspline was least able to capture the seasonal cycle of atmospheric records and therefore we did not consider it further.
We agree that the STL technique has the advantage of allowing flexibility in the shape of the seasonal cycle. Instead, this approach assumes that the seasonal cycle and trend vary only slowly over a defined time window. This assumption is problematic for time-series characterized by rapid or abrupt changes, such as radiocarbon. During the bomb peak, the seasonal cycle is dramatically amplified, and it falls off rapidly in the years that follow. When STL is applied to this time-series, the seasonal amplitude is damped during the bomb peak and amplified in the years that follow compared to observations. Likewise, the bomb peak is damped and delayed in the STL estimate of the trend. This can be partially ameliorated by dividing the time-series into sections, but this then leads to the same kinds of overlap issues that you have highlighted as problematic for CCGCRV.

In addition, we found that the gap filling needed in STL was as problematic for this record as the phase problem in CCGCRV – neither is a perfect choice. We take the reviewer's point that STL doesn't necessarily require gap filling, but this would require an entirely new fitting

system that is not currently used in the atmospheric community and would therefore raise a number of questions of its own.  Further, the seasonal cycle is quite small after 1979, and the majority of users of this dataset are interested in the annual trend, so overall, we judged that the phasing problem in CCGCRV is less problematic than the gap filling problem of STL. We have made some adjustments in the text to clarify these points, but note that we chose not to explicitly discuss HPspline at all since its limitations have been discussed elsewhere already.

**When you investigate the phase change in the 14CO2 signal, you find that the seasonal cycle weakens between 1978 and 1980, and then reverses. Could it be that this timing is related to the change in the fitting sections (1966-1979 and 1980 to1989). The described method for overlap and interpolation between different fits favors the weakening of the seasonal cycle at the section borders if both sections are out of phase. I wonder if you would find the same timing for the phase change if you chose different fitting sections...**
In fact, we chose the division at 1979-1980 precisely because the change in seasonal cycle is apparent in the raw observational data at this time period.  We tested other divisions into time periods and found that the fitted curve couldn't match the data as well, as diagnosed by the residuals.  We have added text to explain our choices.

**Hypothesis of reversed seasonal cycles in the early post-bomb era:**
**The hypothesis behind the changing phase in the seasonal cycle should be backed up by a small (box-) model exercise. This model should include the seasonal cycles of the STE (in NH and SH) and the CEE (cross equator exchange) in the troposphere and the stratosphere. The Mount Pinatubo eruption is a well-studied phenomenon when it comes to stratospheric transport. see e.g. Aquila et al. 2012. They find middle- stratospheric meridional pathways with mixing times of less than a year. The major stratospheric bomb-peak lasted for about 4-5 years (see HASL data compiled in Naegler et al 2006). Can you show in a (box-) model that with those boundary conditions your hypothesis is valid?**
**Aquila, Valentina, et al. "Dispersion of the volcanic sulfate cloud from a Mount Pinatubo–like eruption."** *Journal of Geophysical Research: Atmospheres* **117.D6 (2012).**
Please see our general comment to the editor – we spent considerable effort developing a box model, but ultimately demonstrated for ourselves what previous authors had already shown – $^{14}CO_2$ can't be adequately described with a four box model.  Instead we have revised our discussion of the seasonal cycles to take on the reviewer comments and utilize previous modelling studies.  We have considerably reduced the discussion of the seasonal cycle since the reviewers point out that we don't have sufficient evidence to back it up.

**Interpretation of the seasonal cycles since 2005:**
**I have a couple of questions and comments to the comparison of the Wellington and Cape Grim seasonal cycles:**
**The comparison to the Cape Grim seasonal cycle is problematic since both mean cycles do not average the same time period. Figure 4b shows that there are obvious large inter-annual variations in the amplitude (phasing?) of the seasonal cycle.**
We added a comment that choosing only the period of overlap (2005-2010) gives similar results.

**What is the origin of the double maxima in the BHD cycle?**
We revised the discussion to make our argument clearer that this is due to transport and STE.

**Is the Melbourne influence at Cape Grim detectable in CO2 or CO?**
Yes (in the reference provided) – but the in situ and flask data can be screened to remove the local influences, whereas the $^{14}C$ samples, which reflect the integrated $^{14}C$ signal over ~two weeks, cannot. Text revised to reflect this point.

**Fig 6 does not convince me that BHD is not influenced by anthropogenic emissions. Wellington is in the middle of the "red" area. When reading Pickers et al. they mention that in their data example of the BHD CO2 data they had to gap fil 10% of the data since they deviated from baseline conditions.... To me this indicates some anthropogenic influence at BHD as well.**
We expanded the discussion of influences at Baring Head, using the $CO_2$ observations of Stephens et al (2013) to show that there is a very occasional urban influence, and a more regular terrestrial biosphere influence, but there is no evidence of seasonality in either of these (i.e. they might influence the overall $\Delta^{14}CO_2$ value very slightly, but not the seasonal cycle).

**Sure, Melbourne emits 50 times more ffCO2 than Wellington, however the distance between Melbourne and Cape Grim is 340km, whereas it is around 10km between Wellington and the BHD...**
We revised the text as in the response above.

**If STE is the driving mechanism for the seasonal cycle for the periods 1966 to 1979 and 1980 to 1990, how come that the seasonal cycle post 2005, which is also explained via the STE, is not in phase with the earlier once...**
We have removed this argument since the reviewers have pointed out that we don't have sufficient evidence to back it up.

**Specific comments:**
**p.2 l.40 Please state the years when the measurements in Norway and Austria started**
Done.

**p2. l.44 The term "exchanges" is a bit too general, consider oxidized or something more specific.**
Revised. See also reviewer 3 response.

**p.2 l.45: Production -> Natural production**
Revised.

**p2. L 47: perturbations to Δ14CO2 -> perturbations to natural Δ14CO2 levels**
We considered this, but on rereading the text, believe that "perturbations to $\Delta^{14}CO_2$" more accurately reflects the point we are trying to convey. In recent years, the fossil fuel perturbation is of great interest, but it is the perturbation relative to the recent atmosphere that we are primarily interested in, not the perturbation relative to natural levels.

**p2. L62: Add year to Lopez et al., and add also early attempts of ffCO2 emission estimates like e.g:**
*Meijer, H. A. J., et al. "Isotopic characterisation of anthropogenic CO 2 emissions using isotopic and radiocarbon analysis."* **Physics and Chemistry of the Earth** *21.5 (1996): 483-487.*
*Gamnitzer, U., U. Karstens, B. Kromer, R. E. M. Neubert, H. A. J. Meijer, H. Schroeder, and I. Levin (2006), Carbon monoxide: A quantitative tracer for fossil fuel CO2? J. Geophys. Res., 111, D22302*
We added the year to Lopez et al., and added the Meijer paper. There is now a long list of papers that use $^{14}$C to understand fossil fuel emissions, and it does not seem appropriate to cite every one of them here. Instead, we tried to list only the key papers describing the method and one from each "scale" of study. We should most definitely include the Meijer et al paper, but the Gamnitzer paper doesn't add much beyond the seminal Levin 2003 paper (at least in this context).

**p.2 l77: add citations to the last part of this paragraph**
Done.

**p4 l128: what do you intend with the term "nominally CO2-free"? Did you process blank NaOH solutions? How much CO2 is in a blank NaOH solution? What is the 14C activity of this blank?**
This is in the supplementary material section S3.1 and we added a pointer to that section in the main text.

**p4.l131: "large tray" can you state the surface area of that tray?**
Added in the supplementary materials.

**P4.139: Please add the statement about fractionation (supplement S3.l90-92) to the main text.**
Done.

**P5 l189 "one" sd? In Fig S2 and the text you state 2 sd?**
We have slightly altered the statistical analysis to include a paired sample t test and reworded in both the main text and supplement figure S2 caption to clarify that there is no significant difference between the two methods.

**P6 l259 please include a reference to Fig.2 in this subsection**
Done.

**P8 l316 I don't see the 2005 EN Tandem improvement mentioned in Zondervan et al 2015.... Maybe I overlooked it?**
The method for using all three isotopes measured in the AMS is described in Zondervan et al 2015. We moved the reference to the previous sentence to clarify that it applies to the method, not the improvement in precision.

**P8 l336 Do the measurements from this period carry a special flag (e.g noisy) in the dataset? Reading the supplement I found that you are already doing this. Maybe make a short note in the main text.**
Done.

**P8 l353 how does ccgvu handle data gaps and inconsistent sampling frequencies? Since the paper is (at least for me) not freely available it is worth mentioning this shortly in the supplement.**
Done.

**P9 l362 what is the unit of the cutoff criteria in the frequency domain?**
Days.  This was a typo.

**P9 l363 is the 2 year overlap a good idea? In terms of transition yes, but don't you have now the influence of end-effects in 4 years?**
We tested using different overlap periods.  Using a shorter overlap causes a nasty end effect jump in the record, and a period that is much longer smooths out the differences too much.  We added a comment to justify our choice.

**P9 l368 "mean residual difference" do you mean RMS of the residuals**
No, this is the mean of the residuals, which are the mean difference between the smooth curve fit and the measured values.  We reworded "mean residual difference" to "mean difference", which we think makes this clearer.

**P9 l379 state the "n" of the MC**
Done.

**P9 l382 where are the 95% conf intervals given? In the data set I see only one uncertainty column, please specify in the data-set if this is the 1 sigma error or the 95% conf interval.**
We have removed the 95% confidence interval – we had originally included this in the reported dataset, but since it is effectively multiplying the one-sigma uncertainty by two, it seems unnecessary to include it in the final dataset.

**P9 l384ff the model simulation are not convincingly not used in the paper. See general comments. Consider skipping the subsection 3.7 and Fig 6.**
On lines 479-481 of our original manuscript, we describe the finding of Ziehn et al. [2014] that Cape Grim is influenced by fossil fuel emissions from Melbourne in the wintertime. Ziehn et al show that this seasonal fossil emission influence is primarily driven by seasonal changes in atmospheric transport, rather than seasonality in the fossil fuel emissions.  We present the seasonal analysis of our model simulations to demonstrate that Baring Head is not influences by seasonal transport variability.  We have clarified the text in this section.

**P9 l388 LAU ??**
Removed – this was an oversight as the model also generates footprints for the Lauder site (LAU) that are not discussed in this paper.

**P10 l403ff include ref to fig. 2**
Done.

**P10 l442 30 per mil amplitude for the period 1966-1979? I only see such an amplitude once? A mean amplitude of ca. 7 per mil seem more realistic.**
We revised the text to clarify that 30‰ is the maximum amplitude and a mean across this period of 6 ‰.

**P11 l456 fig 6 -> fig 5??**
Removed this figure reference.

**P11 l459 "Between 1978 and 1980 the seasonal cycle weakened". This is not really seen in fig 4b.**
**Unfortunately 1978 to 1980 is a boundary of the fitting sections... since the seasonal cycles for the two sections are opposed and the overlap is linearly interpolate between fits... a weakening can also come from the applied method.**
In figure 4b, we added the detrended raw observations and expanded the text.

**P11 l460 5 per mil amplitude? Maybe two times in this period... 3 per mil on average**
Revised. (we had used peak to peak amplitudes and have revised to use middle-to-peak amplitudes as is more standard).

**P11  l467    fig 5 -> fig 4**
We intended to refer to figure 5.  No change.

**P12  l 494    fig5 -> fig 4?**
We intended to refer to figure 5.  No change.

**P12  l497    "records that are indicated in figure 1" -> "records where the sampling locations are indicated in figure 1"**
Revised.

**P13  l563   Model results from Levin et al. 2010 already suggest the development of a interhemispheric gradient in the same magnitude for the same time... without changing the southern ocean... although they admit that they are not matching the data...**
Levin et al. (2010) were the first to suggest the development of an interhemispheric gradient, and we were remiss in our discussion of this.  It has been rectified in the revised manuscript.  Levin et al. were able to roughly match the observed gradient without changing the Southern Ocean.  It is important to note that Levin et al. tuned the terrestrial biosphere component of their model to match the observed global average atmospheric $CO_2$ and $\Delta^{14}CO_2$.  Thus, this paper highlights the fact that a terrestrial process occurring predominantly in the Northern Hemisphere can reproduce the observed gradient, but we do not feel that it proves the gradient was caused by the terrestrial biosphere or rules out a major role for the Southern Ocean.
We have almost entirely re-written this section of the paper to present a more balanced view of the potential processes controlling the gradient.  We still feel that a re-organization

of the Southern Ocean is the most likely cause given the supporting evidence from the ocean carbon cycle community. However, we now more clearly acknowledge the previous interpretation of Levin et al. and the fact that we cannot robustly distinguish between a terrestrial and oceanic cause with the existing sparse radiocarbon network.

**Table1: include sample no. to NZ/NZA, replace GC with gas counting, change "measurement methods" to "measurement and sampling methods"**
We made these changes and added more text to the caption to clarify.

**Table2: provide the unit to the 14C differences**
Done.

**Figure1: provide scales to the google earth pictures, indicate urban areas in the upper map.**
Done.

**Figure 2. consider vertical grid lines to illustrate the different periods used in the paper.**
We tried this but found that it cluttered the graph too much.

**Figure 2. Consider indicating graphs with a) and b)**
We think (top) and (bottom) are appropriate here since it is quite obvious what is shown.

**Figure 2. x-label of graph a) is cropped...**
This looks fine in our version. If the problem still appears in the proofs, we will correct it.

**Figure 4. Consider indicating graphs with a) and b)**
We are happy with using top and bottom.

**Figure 4. in a) use the same periods as in the text.**
This was a labelling error and has been corrected.

**Figure 4. b) consider vertical grid lines to illustrate the different periods**
That would be nice, but once we added the detrended observations, adding vertical lines was just too confusing.

**Figure 6. Motivate the plot better. Not really used in the paper. Explain the unit.**
The unit is now more clearly explained, but we chose to keep this figure for reasons outlined above.

**Figure 7. Consider indicating graphs with a) and b)**
As before, we are happy with using top and bottom.

**Figure 7. Consider usage of open symbols. Especially after 2000 it would be good to see all data.**
We tried a number of different ways of presenting this – smaller symbols are hard to see, and open symbols also make it hard to look at. The version we show gave (at least in our opinion) the best presentation of the comparison.

**Supplement:**

**S2.l74   state the surface area of the pyrex tray**
Added.

**extraction follows -> extraction from 1995 onward follows**
Changed.

**in total after flagging you have 427 targets, if you split them between the machines you have 397 and 102 .... To me this does not add up? What am I missing?**
A mistake on our part – we initially recorded the degrees of freedom in each $\chi^2_\nu$ calculation rather than the number of targets (degrees of freedom = number of targets – number of unique samples).  We have rectified this to give the number of targets.

**Please state the main offset for the QC datasets between the two AMS machines.**
We expanded this sentence to say that no offset was observed.

**S5 l217ff What is RLIMS?**
RLIMS is defined in line 36 of the supplement, it is the name of our radiocarbon laboratory database.  We added a reminder at this point in the document since the reader might not recall.

**S6. L262 Indicate the figure S1 with a) and b). I assume a) is Eastbourne and b) is Baring Head? Correct?**
We revised the caption – a is the full record and b is zoomed into the recent time period.

**S9 l394 Since you cannot decide between "red" or "green" for the Baring Head tree, how can you than state the excellent agreement? Is it excellent for both red and green? Please include a link to the t-test or the mean difference to reinforce this statement.**
See above comment – the different colors indicate where we shifted the ring counts by + or – one year NOT different trees.  The bottom graph is simply a zoom of the top one.  We revised the caption to make this clearer.

**S12.l457 Define "NIK".  Why is there only one comparison for NIK and 4 comparisons for BHD?**
NIK has been changed to "Eastbourne" (NIK is the short name for the street the trees are located on).  Most of the Eastbourne samples are from a single tree, and the comparison between the two trees does not appear to be critical (hence only one comparison).  The key comparison is between the BHD and Eastbourne trees, which show no significant differences between $\Delta^{14}CO_2$ at the two locations.

**S12.l468 please specify the t-test: I assume you use a dependent t-test for paired samples? Since the applied formulas are easy it might be clearer if you just explicitly state them.**
Revised the caption.

**what is the mean difference if you use the one year shifted BHD tree (red points in fig S1)?**

Shifting the BHD tree one year older gives a mean difference between the BHD and Eastbourne tree rings of 5.6 ± 0.7 ‰ and a paired sample t value of 8.  Conversely, shifting one year younger gives a mean difference of -8.4 ± 0.8 ‰ and t of 11.  Either shift indicates a poor match and therefore unlikely.  We added some text to describe this.

**Technical comments:**
**In the text please use a consistent ordering (e.g. temporally ascending) when citing multiple papers.**
Done.

**Reviewer 2:**
**The atmospheric radiocarbon measurements conducted at Wellington are a very important record and the authors' efforts to maintain and evaluate the observations are valuable to the community.**
**However, there are some major revisions needed before publication of this manuscript. Much of the paper is used on re-reporting trends and gradients that have already been shown in other work. The authors also make unsupported claims about the mecha- nisms driving the interhemispheric gradient and seasonal cycles of D14C.**

**The paper postulates a sensitivity to Southern Ocean air-sea exchanges that is mis- leading and unsupported. It gives the impression that the Southern Ocean only began influencing the interhemispheric D14C gradient in 2002, whereas the Southern Ocean has always been a primary influence on the interhemispheric D14C gradient, via gross, not net, carbon exchange. Levin et al. 2010 and Randerson 2002 clearly show that the observed trend in the interhemispheric D14C gradient is consistent with a long-term change in the oceanic influence, dominated by the long-term decrease in atmospheric D14C and the change in D14C disequilibrium over the Southern Ocean, which is fur- ther supported by the Graven 2012 papers.**
**A change in upwelling is interesting to consider as a secondary effect, but the authors do not include quantitative models or estimates of how large the effect could be, nor any specifics on how it influences D14C. Furthermore, the Wellington data from 1995-2005 are shown to have serious issues, which would complicate identification of a signal originating in the early 2000s. And there is no discussion about the period in the 1990s when upwelling was increasing.**
We agree with the reviewer that ocean disequilibrium has been important throughout the post-bomb [14]C record, and our text was intended to convey that point, and that there is a possibility of a change in upwelling that could change the magnitude of this effect.  Based on this and comments from the other reviewers, we have reduced the discussion of the possible change in upwelling, and tried to emphasize that indeed ocean exchange has always been important.

**The authors similarly make statements about the influences on the seasonal cycle of D14C at Wellington that aren't well-supported.**
Based on this and the other reviewer's comments, we have shortened the seasonal cycle discussion.

**The paper should be shortened to minimize the re-reporting of previous observations, reduce repetition, clarify the long-term trend in the Southern Ocean influence on the interhemispheric D14C gradient, and remove unsupported statements. As the main contribution is to revise the Wellington data, i.e. no new modeling or other evidence is given to help interpret the data, the paper might be better suited to a journal like Radiocarbon or Atmospheric Measurement Techniques.**

We appreciate the reviewer's view that this work could be well suited to Radiocarbon or AMT. We do believe that the uniqueness of the record, its length and wide use across a large audience makes this worthy of publication in ACP.

**Specific Comments.**

**Section 3.5.3 appears to show major problems in the measurements for the 1995- 2005 period, with large scatter and a high bias. I don't agree that the questionable data should be retained, as the authors have done - "in the absence of better data, we retain both the original and remeasured NaOH sample results in the full record." This conflicts with the aim of the paper to evaluate and refine the previously reported mea- surements and, presumably, to prevent the interpretation of measurement problems as real atmospheric variability.**

We believe that it is appropriate to report these results in the observational dataset, rather than simply discarding them from the published record, since we cannot definitively say that they are wrong. We have flagged them clearly in the dataset, and users have the opportunity to use them or discard them. Further, we provide two different fitted curves – one including this data and the other removing it and replacing with Cape Grim data. We have added text to clarify these points.

**The code WLG is already used by NOAA for Mt Waliguan, China – perhaps another code would be better.**

We have changed the code to BHD, and the actual site (Makara or Baring Head) is still indicated in the data files.

**L15 Earliest direct atmospheric**

Changed.

**L98 Revisiting key findings can be placed in the introduction for brevity.**

We believe that the paper is easier to read with the current organization.

**L104-108 Unsupported. See above comment.**

We removed these sentences from the introduction and shortened the discussion in the results/discussion section.

**L234 Please quote a value for precision**

Added.

**L306 Why would this result in higher D14CO2?**

Revised to "This would result in contaminating $CO_2$ absorbed on the NaOH before the solution was prepared. Since atmospheric $\Delta^{14}CO_2$ is declining, this would result in higher $\Delta^{14}CO_2$ observed in these samples than in the ambient air. "

**L378 More detail needed. Where is this used?**

We added the following sentence to clarify why this is included: "This is provided for further users of the dataset, and may be particularly helpful when the dataset is used for aging of recent materials."

**L384 How do 4-day back trajectories address the seasonal cycle? The panels in the figure all look the same. This is not very useful. A panel should be shown with the differences if there is a difference to highlight.**

Ziehn et al. (2014) show that the Cape Grim site is influenced by seasonally coherent changes in the atmospheric transport, such that the site detects fossil fuel emissions from Melbourne in winter but not in other seasons.  We show these model simulations precisely to demonstrate that the Baring Head record is not influenced by such seasonal variations in transport.  In response to this comment and a similar comment by the first reviewer, we have rewritten and clarified this discussion in the manuscript.

**L413 Since 2005 or earlier?**

Changed "since" to "after" to clarify.

**Section 4.1 seems out of place and repetitive. Should move to introduction and focus on new results here.**

We believe that the paper reads more clearly with this discussion here.

**L435 Turnbull 2009 only includes simulations from the 2000s, so they do not show the Suess Effect became the dominant driver in the 1990s.**

We included references to the two studies that have shown that the Suess Effect is the most important driver after 1990.  Levin et al 2010 show this has occurred since 1990, Turnbull 2009 is a second study using an independent model that agrees with the Levin result.  We believe it is appropriate to include both references.

**L454 Do you mean when mixing with lower-D14C air from the stratosphere was the strongest? Are there Southern Hemisphere stratospheric observations from the bomb period supporting the idea that tropospheric D14C was higher than stratospheric D14C? Are you saying that tropospheric D14C was higher than stratospheric D14C in the Southern Hemisphere until the late 1970s? Bomb 14C would have also entered the SH stratosphere through the tropical tropopause, while at the same time tropo- spheric D14C was declining, so this seems unlikely. Note Northern Hemisphere sites also showed minima in spring in the early bomb period. Levin 2010 simulate recent seasonal influences on D14C and should be cited here. Oceanic influences on the seasonal cycle should also be mentioned.**

We have revised this section to remove this discussion.

**L468 See Brenninkmeijer, C. A. M., Lowe, D. C., Manning, M. R., Sparks, R. J., & van Velthoven, P. F. J. (1995). The 13C, 14C, and 18O isotopic composition of CO, CH4, and CO2 in the higher southern latitudes lower stratosphere. Journal of Geophysical Research: Atmospheres, 100(D12), 26163-26172. doi:10.1029/95JD02528**

Thank you for this reference, but we have removed this discussion and therefore not included it.

**L494 This is the time of maximum in the NH so this phasing is unexpected. Is there an explanation for the double-peaked shape of the cycle? This section relies on dismissing the Cape Grim data, which is not entirely convincing. Are other Southern Hemisphere observations relevant here?**
There are no other long term records from a similar latitude in the Southern Hemisphere (there are tree ring records, but these clearly cannot resolve seasonal cycles). We expanded this discussion to strengthen our argument. It is worth noting that the seasonal cycle during this period is quite small and the difference between the seasonal cycle in the two records is perhaps 0.5‰.

**L517 It would be useful to include a plot of the difference between the Wellington and Cape Grim data.**
The two datasets are shown in figure 3 and we added a reference to figure 3 in this sentence.

**L521 Delete the word signal. Is it possible to say something more quantitative than "homogeneous"?**
In the previous sentence, we say that differences between the two sites are smaller than the measurement uncertainty.

**L527 What is the basis for the new estimate of the interhemispheric exchange time? How was this calculated? Without any supporting information this paragraph should be deleted.**
We have added further explanation of this calculation. It is surprising that this bomb peak difference has never actually been used to calculate an interhemispheric exchange time before. Although our calculation is simplistic, it agrees nicely with recent, more sophisticated analyses of the exchange time and we think it is worth including.

**L544 Need to cite Levin 2010, and Graven 2012**
Both are now cited in this paragraph.

**L561 Also shown in Randerson 2002 and Levin 2010**
We now include the Levin 2010 reference. Randerson 2002 doesn't go beyond 2000 in its data, so it is less relevant here.

**L565 This paragraph is misleading. See main comment above.**
We have shortened this section considerably.

**L575 This is the gross carbon flux not the net carbon flux. Atmospheric D14C has been highly sensitive to Southern Ocean upwelling not only since the 1980s but since the preindustrial period and throughout the bomb peak period – see Randerson 2002 and Levin 2010**
We agree with the reviewer, see earlier comments. And have shortened this section considerably.

**L593** "Although the changing Southern Ocean carbon sink is the most likely explana- C4 tion," Atmospheric D14C is not directly affected by the Southern Ocean carbon sink. What is the justification for this statement? See main comment above.

See previous comments – shortened this section.

**Reviewer 3 J. Miller (Referee)**

**General comments.**

**This paper documents and analyzes the longest atmospheric radiocarbon time series from a single site. Obtained near Wellington, New Zealand starting in 1954 and con- tinuing to the present, these data represent a signature time series of carbon cycle science. The authors document the revision and evaluation of the data, which should lead to a significant improvement in its scientific utility. The seasonal cycle and trend are analyzed convincingly, although too much attention is paid to the hypothesis that an increased Southern Ocean CO2 sink can explain the changing Δ14C atmospheric north-south gradient. While it's true that the change in the north-south 14C gradient supports this idea, there is no new analysis of the time series to bolster it. One ad- ditional point is that it would be good to provide the internet location of the data in addition to the static spreadsheet provided. Presumably the ftp site would contain the data set of record including the latest data, flags, and corrections. Nonetheless, this is a strong paper that is entirely appropriate for ACP; it should be published after a few modifications.**

The dataset is now available at the WDCCGG and our own websites and we have added a section 7 Data Availability at the end of the text with the links. We are working on also putting the data at CDIAC where much of the global $^{14}CO_2$ data resides, but internal CDIAC issues have slowed this down.

**Below, I list some edits and comments by line number.**

**Specific comments.**

**L21,22. While Cape Grim air samples may contain anthropogenic signals in winter, air samples have often been collected during times when the wind is not coming from the north.**

This is not the case for $^{14}C$ samples which are integrated over ~2 weeks.  We have clarified this key point in the text of our paper.

**L44. 'exchanges' is a bit vague. Why not spell it out to say that 14C reacts immediately with O2 to form 14CO, which is subsequently oxidized to 14CO2**

Done.

**L68-70. This is redundant with text around L44.**

The slight repetition seems necessary for the text to be clear.  No changes made.

**L75. Perhaps strike 'now', and add 'in the two decades following the atm. test ban treaty' at the end of the sentence.**
Done.

**L77. I don't agree that the additions of fossil fuels became the dominant factor influencing the 14CO2 trend. If fossil fuel CO2 additions are 'dominant' I would think of them being an order of magnitude or so larger than other processes. Presently (and more or less in the 1990s), fossil fuel combustion alone would reduce the atmospheric Δ14C by ~ 10 per mil/yr; cosmogenic production would increase it by 5 per mil/yr; the land- atmosphere and ocean-atmosphere disequilibrium fluxes would be roughly +4 and -4 per mil/yr. It might be reasonable to try and calculate a point at which the negative trend in atmospheric Δ14C was driven more by fossil fuel emissions than by absorp- tion of bomb 14C atoms into the biosphere and oceans. But this would not equate to 'dominant' in my opinion.**
This is an important distinction, and we agree with your points.  We have changed from "dominant" to "the largest contributor to the $\Delta^{14}CO_2$ trend."

**L80. Change 'especial' to 'special'**
This is a New Zealand colloquialism.  Changed to standard English.

**L129. Use 'M' (molar) or 'mol/L'**
Done.

**L158. 'Faithfully' record Δ14C, but not the 14C content, which is offset by ~ 34 per mil.**
Corrected from "$^{14}C$ content" to "$\Delta^{14}C$"

**L210. Was testing done do see if the samples could be stored for up to three years before analysis without introducing artifacts.**
No such testing has been done, and this is something we will consider for future updates of the record. No changes made to the text.

**L216-218. Could using an offline δ13C value produce bias or just add noise? Any tests to examine this?**
Yes, this is possible, even likely. We have not done specific tests, but fractionation during sample preparation will almost certainly always go in the same direction.  The most likely culprit is incomplete graphitization (in the LG1 graphite system used at this time, reaction completion was not directly measured and we suspect that graphitization was often incomplete), which fractionates to the lighter isotopes and if not diagnosed would result in a higher $\Delta^{14}C$ (i.e.  goes in the direction of the apparent bias in the data).  On the other hand, fractionation in the AMS (most likely in the ion source) is likely to vary in sign through time.  We have added explanation in sections 3.3 and 3.5.3 to explain this more clearly.

**L227. Considering that the multi-target averaging resulted in differences of up to 5 per mil, I think that this deserves a detailed explanation, at the very least in the supplement.**

We agree.  An explanation was already given in the supplementary material and we have expanded it slightly and included a note in the main text pointing to the supplement for more information.

**L243. S+P's Δ is the same as the presently used Δ14C; their Δ14C is defined differ- ently.**
Reworded to clarify.

**L255. How was the weighting done? Inverse square of the measurement precision?**
Weighted mean as defined by Bevington and Robinson (2003).  Sum($x_i$*$w_i$)/sum($w_i$), where $x_i$ is the mean of each measurement I and $w_i$ is the weighting, defined 1/$w_i$.  Since measurement precision does vary, it is appropriate to use a weighted mean rather than a simple mean.

**L280. Wondering if 'excursion' is the best word here. Anomaly?**
Changed.

**L283. As mentioned in comments on L22, Cape Grim sampling can be 'tuned' just for a clean air sector. If the issue is integrated sampling, then I would say that.**
Point taken, but rereading this section, the sampling regime at Cape Grim is not germane in this paragraph (although it is relevant elsewhere in the manuscript).  No changes made.

**L284. Change 'terrestrial' to 'mainland'?**
Done.

**L303-304. 'preparation was conducted' to 'was prepared'.**
Done.

**L313. 'or thereafter' to 'and thereafter'**
Done.  We noticed that one even before the reviewer did.

**L325. I don't see the reduction of scatter shown in any plot. It would be useful to show how the reprocessing improved the noise.**
The data was not reprocessed, it is that once the change was made, the $\Delta^{14}CO_2$ record immediately becomes less noisy.  It is clearly apparent in figure 2b.  We added some words in the text to point the reader to the figure.

**L351. Change 'ccgvu' to 'ccgcrv' which is the actual name of the curve fitting code.**
Done.

**L362. Insert 'day' after 80. Good that this important detail was included.**
Done.

**L395. Add a sentence explaining what a footprint is.**
Done.

**L403. I think 'roughly "natural"' can be deleted; natural is ambiguous. Maybe 'roughly pre-industrial'?**
Done.

**L421-422. By 'long-term' to you mean decline since the 1960s? For many in the radiocarbon world, that wouldn't be very long, so maybe define the time period more explicitly. Also, insert 'known' prior to 'long-term trend in. . .'**
Revised.

**L434. As mentioned earlier, I don't think 'dominant' can be justified.**
Changed "dominant" to "largest"

**L469. I'm wondering about the value of an untestable hypothesis. What you say sounds plausible, but maybe refer to it as speculation?**
We have removed this argument based on reviewer skepticism.

**L507. Should Levin et al reference by 2010? 2013 paper appears to deal with Europe.**
This is correct in the text – we are referencing the method by which the Jungfraujoch (European) measurements are made.

**L527-534. I would like to see the math of how this was calculated, at least in the supplement. Also, one important factor is to know the state of ENSO during the 1963-1965 period, because La Nina, for example, can significantly increase inter-hemispheric exchange. Finally, the SF6 derived value is based purely on surface data, whereas the Δ14C method has a significant upper atmosphere component. It would be good to comment on how the estimates might differ.**
This is a very simple calculation – what is the temporal offset between the first maximum of the bomb peak in each hemisphere. We have revised the text to clarify how the calculation was done.

**L544 – 596. I felt that the text at the end of the Results and Discussion section focus- ing on the interhemispheric gradient and the Southern Ocean was a bit out of place. The Wellington Δ14C data confirm the gradient observed earlier and extend it in time. However, at present, the two paragraphs (starting at line 565) sound more like a review of the Southern Ocean uptake hypothesis, because there doesn't appear to be any new analysis. If it's not possible to add any new analysis using the Wellington data, I think it would be better to be very concise, essentially saying something like 'our data suggest the S.O sink continues to explain. . . Numerous recent studies using methods x, y and z further support. . . Our data set will be a powerful constraint to understanding the evolution of the gradient in a quantitative model framework. . .'**
We have reduced this section to a few sentences. Our intent is to alert readers to the opportunity that Southern hemisphere $^{14}CO_2$ observations give to understanding Southern Ocean carbon cycling.

**L571. Change 'natural' to 'mass-dependent'?**
Done.

**L650. Perhaps acknowledge Scott Lehman and Ingeborg Levin for providing unpub- lished data.**
Acknowledgement added. Although we use only published data, they still generously provided the datasets for us to use.

**Table 2. WLG is already taken as a site code (for Mt. Waliguan Observatory, China), at least with respect to the WMO GAW program. Wouldn't MAK and BHD work here?**
We have changed to use BHD for the overall site code, recognizing that the early part of the record is actually from Makara. However, we want to keep a consistent overall site code so that users are not forced to stitch the two sites together themselves. Another reviewer

raised the same comment.

**Figure 2. Can you distinguish the symbols and/or colors for the two versions of the EN-Tandem: i.e. 12,13,14 vs. 13,14, since the results seemed to be significantly different.**
Done.

---

## Author Response (AR2)

**1 **Turnbull et al: Response to reviewers on revised manuscript**

Comments from the editor and reviewers are in bold type, our responses are in plain type.
 The marked up manuscript and tables/figures are included in this document following the
 point-by-point response.

**6 Editor: I have now considered the revised manuscript and all three reviewers. It is**

7 clear that the manuscript is much improved as a result of the revisions. However,

- 8 the reviewers are still split in their opinions though noticeably less so than before. I
- 9 would like you to address the points the reviewers raise, especially those by
- 10 reviewers 2 and 3. If you wish to send a point-by-point response before revising the
- paper, that will be fine. If not, please submit your response with a revisedmanuscript.
- 12 ma 13
- 14 To give you some guidance, I think some restructuring would make the paper much
- 15 more readable. Section 3.5 could become a new section (4. Data validation), and 3.6
- 16 could form part of the new section 5.2 (Seasonal variability) which should lead to
- 17 some reduction in overall length. I sympathise with reviewer 2's comment about the
- 18 balance (and redundancy) between the main manuscript and the supplementary
- 19 information. I would think that much of the curve fitting text would be better in the
- supplementary information as it underlies, rather than adds to, the main findings. It
- 21 would make the paper more accessible to the non-specialist such as myself.
- 22 We changed Section 3.5 and it's subsections (3.5.1, 3.5.2 and 3.5.3) to Section 4 Data
- 23 Validation with Subsections 4.1, 4.2 and 4.3 as suggested.
- 24 Moving section 3.6 Smooth Curve Fit to the results section was problematic, since we
- 25 discuss the smooth curve in (renamed) Section 4 Data Validation. Therefore, we left this
- 26 section in the methods (renumbered as section 3.5), but substantially shortened the main
- text and moved the rest to the supplementary material (supplementary section 5.4). We
- also made some minor changes in (renumbered) Section 5.2 Seasonal Variability in the
- 29 Wellington Record to be consistent with the changes in the methods sections.
- 30

31 I also sympathise with reviewer 2's comment about Section 3.7 and the use of

- 32 NAME as you do not refer to the analysis much in the text. I suggest zooming in on
- 33 Figure 5. That could make it easier to assess the possibility of local contamination. If
- 34 the suggested calculation can be done, that would be good. However I say that not
- 35 knowing how hard it is to do.
- 36 We removed this section we had considered doing so after the first round of reviews,
- and with these new reviews, we decided to indeed remove it. As for the suggested
- calculation, i is reasonably complex and time consuming, and given that the influence is
- 39 quite small, we felt it was beyond the scope of this paper to add that analysis.
- 40

**41 The other comments are more detailed and I look forward to your addressing them**

- 42 either in a revised manuscript or in your response.
- 43

**44 Reviewer 1 (Samuel Hammer):**

45 No changes suggested or made.

**Formatted: Font:Bold**

- 46
- 47 Reviewer 3 (John Miller):
- 48 a) Regarding gap-filling of time series for ccgcrv, there is a web site describing
   49 ccgcrv where one can also download all the code.
- 50 www.esrl.noaa.gov/gmd/ccgg/mbl/crvfit/
- 51 Thank you, we added a link to the website in the text. 52
- b) The hypothesis that a change in the land carbon sink could explain changes in the D14C latitude gradient is unrealistic considering that D14C is a quantity designed to factor out net fluxes from the land biosphere. line 647 of new pdf

56 We agree, and have added a statement to this sentence "although this latter is less likely

since  $\Delta^{14}CO_2$  is much less sensitive to biospheric fluxes than to either ocean or fossil fuel fluxes (e.g. Levin et al., 2010; Turnbull et al., 2009)."

59

**60 c) WLG is still being used as a site code in Fig. 4.**

61 Fixed. Thank you.62

**63 Reviewer 2:**

64 We especially thank this reviewer for a very thorough review of the paper and some very 65 insightful comments. We hope that the changes we have made have captured the

- 66 reviewer's intent.
- 67
- 68 The authors have made some improvements and removed some unsupported
- 69 statements in the revision. However, the abstract and conclusions are unchanged
- 70 from the original version. These both include the misleading and unsupported
- 71 statements about the Southern Ocean influence on the interhemispheric gradient
- noted in the first review, and they both need to be revised to remove these and to
- 73 focus on the new results reported here.
- 74 We revised the abstract and conclusions to include the alternative (although less likely)
- 75 explanations for the developing interhemispheric gradient.
- 76
- 77 The paper has a strange organization that mixes results (for comparison and
- 78 validation of measurements) into the Methods section, and the Results and
- 79 Discussion section is dominated by discussion (most of the data has been presented
- 80 before and there is no new analysis of the causes of observed cycles, trend and
- 81 gradient). The authors should revise the organization to have a Results section
- 82 which is clearly showing new results from this analysis including the comparison
- 83 and validation of measurements.

We reorganized the sections following the editor's recommendations – see comments
 above.

- 86
- 87 The Results section should clearly report revisions to the Wellington record that
- 88 were made since Currie 2009 and new observations of the BHD D14C trend over
- 89 2005-14 (an interesting new result that is not currently included in the paper), in
- 90 addition to the analysis of seasonal cycles and estimates of interhemispheric
- 91 gradients with other stations already included.

- 92 We added a paragraph to section 5.1 discussing the recent BHD trend, and included a
- 93 calculation showing that the Suess Effect is increasing and cannot explain the slowing in
- 94 the BHD trend.
- 95

**96 Then the Results and Discussion section should be renamed Discussion, and**

- 97 shortened.
- 98 We have followed the editor's suggestions for reorganization (see above).
- 99

**100 Section 3.7: I still find this analysis and figure to not be very useful. Why don't the**

- authors use the footprints to calculate fossil fuel CO2 from local combustion to 101
- 102 make this point more clearly? In any case, please note here that the model
- 103 simulations are only used to detect local influences of fossil fuel combustion and
- 104 potential seasonality
- 105 We thought this through, and ultimately agreed that removing this information and figure

106 was the best choice. Instead we have simply referenced Steinkamp et al., 2017, where the

- BHD footprints are discussed in some detail. We did consider using the footprints to 107 108
- calculate the fossil fuel CO2 from local combustion, but this is a fairly major undertaking 109 since it requires spatially and temporally explicit fossil fuel emissions for New Zealand.
- 110 Since the influence is small, this appears to us excessive for this paper.
- 111

112 Section 4.2: This section is too long and it does not give a clear impression of the

- 113 drivers of the seasonal cycle. Following the recommendation above for
- 114 reorganization, the seasonal cycle results should be separated from the discussion,
- 115 which can be shorted to a couple of paragraphs.
- 116 We extended this section in the first round of revisions, specifically to address questions
- 117 that this reviewer raised. Shortening this section would necessitate removing this added
- 118 discussion and we not done so.
- 119
- 120 Ocean exchange needs to be mentioned as an influence. The authors do not mention 121 the analysis of Levin 2010 on seasonality at CGO, who found stratosphere-
- 122
- troposphere exchange and oceanic exchange were the main contributors.
- 123 Levin et al 2010 argued that seasonal variability in transport (cross-equator and strat-trop) 124 were the main drivers and this is already stated in the text. 125
- 126 Also, Randerson 2002 Fig 6 for the 1980s.
- 127 We discuss the 1980s period and compare with the Randerson results in this section
- 128 already. No changes.
- 129

130 The authors should be careful not to over-interpret the data in this section. Are the

131 authors stating that the biosphere is the primary contributor to the seasonal cycle at

BHD in the 1980s-90s when they say "This is consistent with a change in sign of the 132

terrestrial biosphere contribution as the bomb 14C pulse began to return to the 133

134 atmosphere from the biosphere (Randerson et al., 2002)"? Why would the biosphere

- 135 contribution be strongest in winter?
- We added some words to explain that the seasonality in atmospheric transport convolved 136
- 137 with the flip in biosphere seasonality could drive the change.

**139 L471 Also shown in Graven et al. 2012, La Jolla trend**

- 140 Added this reference.
- 141

**142 L578-580 It's strange to introduce Cape Grim here, when it has already been**

143 referred to several times and compared to BHD data in the previous section.

144 We added the Cape Grim info where Cape Grim is first discussed in the paper. We also

- 145 left it here to help the reader follow where all the sites are.
- 146
* * *
- 149 offsets between the three laboratories?
- 150 We have opted to leave this section where it is, rather than adding another section in the 151 methods. We did consider laboratory offsets and differences, and this is discussed 152 briefly.
- 152 153

154 L593-601 This is not a new or robust estimate of the exchange time and needs to be 155 deleted. Many studies have noted the delay of about a year between the peak D14C

- in the two hemispheres, e.g. Nydal 1966, with Lal and Rama 1966 providing a more rigorous estimate than offered here.
- 158 In preparing this paper, we were surprised how little published information there actually
- 159 is on the use of the 14C bomb spike to look at interhemispheric exchange time, since it is
- 160 "common knowledge" that this is one of the applications of bomb 14C measurements.
- 161 Following this comment, we re-read those early papers, and they estimated exchange
- 162 times of about 1.7 years. In neither case did they include data beyond 1964, which means 163 that they did not capture the appearance of the bomb  $^{14}$ C peak in the Southern
- 164 Hemisphere.
- 164 Heim 165

**166 L633 Levin et al. 2010 also attribute the change to the (Southern) Ocean**

- 167 Agreed, and re-reading the text, we feel this is already stated sufficiently.
- 168

169 L636 Where is it shown that the interhemispheric gradient develops rapidly? From

- 170 Levin 2010, Fig 6i there appears to be a clear trend over 1985-2005, albeit with
- 171 interannual variation. No such figure is included here.
- 172 We added a reference to figure 5, where this is shown. Note also that the development
- and continuation of this gradient is clearer with the longer records shown here than in
- 174 either the Graven or Levin papers.
- 175
- 176 L638-651 should be deleted and replaced with one sentence such as "The
- 177 interhemispheric gradient may also be influenced by the apparent reorganization of
- 178 Southern Ocean carbon exchange in the early 2000s (Landschützer et al., 2015),
- 179 which is postulated to be associated with changes in upwelling of deep water
- 180 (DeVries et al., 2017) to which atmospheric Δ14CO2 is highly sensitive (Rodgers et
- 181 al., 2011; Graven et al., 2012b)."
- 182 We disagree on this point. The most likely explanation for the interhemispheric  $^{14}$ C
- 183 gradient is Southern Ocean upwelling, and there is wide interest in the possibilities of

- 184 using 14C to understand Southern Ocean carbon cycling. Thus we think it deserves more
- 185 than one sentence to explain the rationale.
- 186

**187 For "aging", it may be better to say "dating"**

- 188 We think the reviewer is referring to "aging" of recent materials. We have retained the
- 189 word "aging" since "dating" is not normally used for post-bomb radiocarbon age
- 190 determination.
- 191
- 192 The authors argue for retaining the data with large scatter over 1995-2005, but
- 193 these data flagged "T" and the smoothed record over this period need to at least
- 194 include larger uncertainties. The two smoothed records provided "BHD" and
- 195 **"BHDCGO"** differ by up to 10 per mil, but the reported uncertainties are about 2
- 196 per mil. Shouldn't the RMSE of 5 per mil over this period be incorporated in the
- 197 uncertainty of the smoothed record of BHDCGO?
- 198 There are many possible strategies for evaluating uncertainties. Our view is that for this
- 199 work it is most appropriate to report uncertainties based on the dataset itself, rather than 200 attempting to incorporate biases diagnosed from other datasets. The atmospheric
- 201 radiocarbon community is in the process of strengthening intercomparisons between
- 202 laboratories and developing harmonized global and regional datasets. Uncertainty
- analysis from comparison of multiple records such as that suggested here will be
- 204 incorporated into that future work.
- 205

**206 Can more information be provided on the "additional error term, determined from**

- 207 the long-term repeatability of secondary standard materials"? The Turnbull 2016
- 208 reference appears to report only recent secondary standard measurements since
- **209 2011.**
- 210 This additional error term has been added to measurements for all Rafter AMS
- 211 measurements, but the methodology had never been published prior to the Turnbull et al
- 212 2015 paper. A longer discussion is included in the supplementary material and we added
- a reference to that in the main text.
- 215 The excel spreadsheet refers to WLG in the header text on several sheets.
- Fixed. Thank you.
- 217

**218 It seems unnecessary to repeat so much text from the paper in the supplement.**

- 219 The supplement includes more detail of the dataset than is in the paper, and we feel that it
- 220 becomes difficult to follow if users need to jump between the main paper and the
- supplement to get all the information, therefore we prefer to include all the detail in thesupplement.
- 223

**Supplement does not define "trend" and "fit" reported in the spreadsheet. "Trend" suggests a rate of change rather than deseasonalized data.**

- 226 We moved the bulk of the information on the smooth curve fits from the main paper into
- 227 the supplementary material and added into comments there to indicate how "trend" and
- 228 "fit" are defined in the datasets.
- 229

**Sixty years of radiocarbon dioxide measurements at Wellington, New Zealand 1954 - 2014 231**

Jocelyn C. Turnbull1,2,\* Sara E. Mikaloff Fletcher3, India Ansell1, Gordon Brailsford3, 233

234 Rowena Moss3, Margaret Norris1, Kay Steinkamp3

235

1GNS Science, Rafter Radiocarbon Laboratory, Lower Hutt, New Zealand 236

- 2CIRES, University of Colorado at Boulder, Boulder, Colorado, USA 237
- 3NIWA, Wellington, New Zealand 238
- 239 \* contact author: j.turnbull@gns.cri.nz

**240 1. Abstract**

- We present 60 years of  $\Delta^{14}$ CO2 measurements from Wellington, New Zealand (41°S, 241
- 242 175°E). The record has been extended and fully revised. New measurements have been

243 used to evaluate the existing record and to replace original measurements where

- 244 warranted. This is the earliest direct atmospheric  $\Delta^{14}$ CO2 record and records the rise of
- the 14C "bomb spike", the subsequent decline in  $\Delta^{14}$ CO2 as bomb 14C 
[revised manuscript text omitted]
 bicker $\Delta^{14}CO_2$ absorbed in these complex then in the curling of the second |     |                                 |  |
| 282
594 | night $\Delta$ UO 2 observed in these samples than in the ambient air. Another possibility is that there were known issues with the heak around contamination in the proposition $[1]$                                                                                                                                                                                                                                                                                                                                                                                                                                                                                                                                                                                                                                                                                                                                                                                                                                                                                                                                                                                                                                                                                                                                                                                                                                                                                                                                                                                                                                                                                                                                                                                                                                                                                                                                                                                                                                                                                                                                      |     |                                 |  |
| 200        | that there were known issues with the background contamination in the proportional                                                                                                                                                                                                                                                                                                                                                                                                                                                                                                                                                                                                                                                                                                                                                                                                                                                                                                                                                                                                                                                                                                                                                                                                                                                                                                                                                                                                                                                                                                                                                                                                                                                                                                                                                                                                                                                                                                                                                                                                                                                     |     |                                 |  |

- counters during this period that could result in a high bias  $\Delta^{14}$ CO2. In any case, these
- 590 values are anomalous and we remove the original NaOH static sample measurements
- 591 between 1990 and 1993 and replace them with the new flask measurements for the same
- 592 period.
- 593

**594 **4**.3. 1995-2005 variability**

- 595 As already discussed in section 3.3, the measurement method was changed from gas
- 596 counting to AMS for samples collected in 1995 and thereafter. During the first ten years
- of AMS measurements, the record is much noisier than during any other period (figure 2). Until 2005, offline  $\delta^{13}$ C measurements on the evolved CO2 were used in the
- normalization correction. In 2005, online  $^{12}$ C measurement was added to the AMS
- system, allowing online AMS measurement of the  $\delta^{13}$ C value and accounting for any
- fractionation during sample preparation and AMS measurement (Zondervan et al., 2015;
- see also section 3.3). This substantially improved the measurement accuracy and the
- 603 noise in the  $\Delta^{14}$ CO2 record immediately reduced as can be seen in the lower panel of
- figure 2. Therefore, we suspect that the variability and apparent high bias in the 1995-
- 2005 period of the  $\Delta^{14}$ CO2 record is due to measurement uncertainty and bias rather than atmospheric variability.
- 607
- 608 The remaining NaOH solution for all samples collected since 1995 has been archived,
- and typically only every second sample collected was measured, with the remainder
- 610 archived without extraction. In 2011-2016, we revisited the 1995-2005 period,
- 611 remeasuring some samples that had previously been measured and some that had never
- 612 been measured for a total of 52 new analyses.
- 613
- 614 The new measurements for this time period do show reduced scatter over the original
- analyses, particularly for the period from 1998-2001 where the original analyses appear
- anomalously low and in 2002-2003 when the original analyses appear anomalously high.
- 617 Yet there remain a number of both low and high outliers in the new measurements.
- These are present in both the samples that were remeasured and in those for which this
- 619 was the first extraction of the sample. This suggests that a subset of the archived sample
- 620 bottles were either contaminated at the time of collection, or that some bottles were
- insufficiently sealed, causing contamination with more recent CO2 during storage.
   Comparison with the tree ring measurements and with the Cape Grim record (Levin et al.,
- 2010) suggest that the measurements during this period may, on average, be biased high
- as well as having additional scatter (figure 3). Nonetheless, in the absence of better data,
- we retain both the original and remeasured NaOH sample results in the full Wellington
- record, with a special flag to allow users to easily remove the questionable results if they
- 627 prefer. We also provide a smoothed fit that excludes these data (section 3.6).
- 628

**629 **5**. Results and Discussion**

- 5.1. Variability in the Wellington record through time
- 631 The Wellington  $\Delta^{14}$ CO2 record begins in December 1954, at a roughly pre-industrial
- 632  $\Delta^{14}$ CO2 level of -20 ‰ (figure 2). From 1955,  $\Delta^{14}$ CO2 increased rapidly, near doubling

[revised manuscript text omitted]